# UNISE: Unified Noise-Invariant Learning for Speech Enhancement Toward Improved Content Preservation

## Abstract

The importance of semantic information in speech enhancement (SE) has recently been emphasized to improve intelligibility, whereas earlier work primarily focused solely on acoustic perceptual quality. To address this, recent approaches leverage pre-trained self-supervised representations, which have shown strong performance on discriminative tasks. However, such representations are less effective for generative tasks and, since they are typically trained only on clean data, struggle to fully preserve content under noisy or distorted conditions. In this work, we aim to bridge this gap by introducing a unified generative SE model, called **UNISE**, that incorporates noise-invariant representation learning. By jointly learning an encoder using noise-invariant clustering and a generative decoder, our model produces robust speech representations well suited for the SE task. As a result, UNISE achieves improved linguistic content preservation while maintaining competitive perceptual quality[1].

## 1 Introduction

Speech enhancement (SE) aims to improve the intelligibility and perceptual quality of speech signals degraded by, e.g., additive noise or reverberation. Most previous SE methods have been *predictive* (also referred to as *discriminative*), learning a deterministic mapping from noisy to clean speech (Luo & Mesgarani, 2019; Défossez et al., 2020). However, they often struggle to generalize and fail to recover missing spectral details. To overcome these limitations, recent works have explored *generative* models for SE (Lu et al., 2022; Serrà et al., 2022; Richter et al., 2023; Jukić et al., 2024), which provide improved performance and deliver higher perceptual quality.

While generative approaches have shown promise, they often *hallucinate*, e.g., by changing linguistic content or speaker characteristics, because they are trained solely on *acoustic* criteria. This phenomenon has also been observed in Saijo et al. (2025), which reported occasional hallucination of spoken content specifically in generative SE models. To mitigate this, several SE methods have incorporated pre-trained speech representations from self-supervised learning (SSL) to enrich *semantic* information (Yang et al., 2024; Guimarães et al., 2025; Wang et al., 2024; Yao et al., 2025), which substantially improves both content accuracy.

Most existing SSL speech representations (Baevski et al., 2020; Hsu et al., 2021; Baevski et al., 2022; Liu et al., 2023) have emerged as foundation models for various downstream discriminative tasks such as automatic speech recognition (ASR) and speaker identification. Ideally, they could also be used for the estimation of full speech signals, e.g., generative SE or speech separation. However, the performance of SSL representations in such tasks, when used alone, remains limited (Tsai et al., 2022) since their training primarily focuses on capturing phonetic and semantic content rather than the fine acoustic details required for generation. To address this, UniWav (Liu et al., 2025) proposed a unified pre-training framework targeting both discriminative and generative tasks.

Furthermore, most existing speech representation models are trained exclusively on clean speech, and only a few prior works (Chen et al., 2022; Zhu et al., 2023; Ng et al., 2023; Chang & Glass, 2024) have investigated their robustness under noisy or distorted conditions. However, the robustness of

---

[1]Audio samples are available at: `https://tinyurl.com/UNISE-ICLR2026`

these upstream SSL models remains a critical bottleneck for SE. SSL models trained exclusively on clean speech lose significant content information when faced with noisy inputs. While models like WavLM (Chen et al., 2022), which are pre-trained with noise augmentation, demonstrate improved robustness, they still suffer from substantial information degradation as noise levels increase.

To summarize, in the context of SE using speech representations, the challenge is twofold:

1. The representation must be robust to noise and distortions.

2. The representation must retain sufficient generative capability to guide the generation of high-quality waveforms.

To bridge this gap in existing methods that rely on common speech representations, we propose **UNISE**, a **U**nified **N**oise-**I**nvariant learning for **S**peech **E**nhancement, which jointly performs generative speech enhancement training and robust noise-invariant representation learning.

To tackle the first challenge (robustness to degradations), Babaev et al. (2024) suggested two rules: the (i) clustering rule, where speech signals that "sound identical" should always form a single cluster, and the (ii) SNR rule, where decreasing signal-to-noise ratio (SNR) should monotonically shift representations away from that cluster. Ideally, robust representations would satisfy the clustering rule even under noise; however, since common SSL representations cannot achieve this, they instead rely on the SNR rule as a fallback. By contrast, our noise-invariant learning explicitly enforces disentangled, noise-robust clustering, naturally capturing SNR variation without resorting to this compromise. By explicitly modeling diverse noises through noise-invariant training, UNISE ensures that the learned representations remain stable across various distortions and effectively preserve the underlying content.

For the second challenge (generative capability), we follow UniWav (Liu et al., 2025), which integrates a representation encoder and a generative audio decoder. Building on the joint training of the encoder and decoder, we extend the framework to support noise-robust representations for both discriminative and generative tasks. The joint reconstruction training guides the encoder's representations to support the reconstruction task during training, further improving the noise robustness of the representation from the encoder (Wang et al., 2022; Guimarães et al., 2023) and strengthening its generative capabilities. UNISE integrates the encoder with noise-invariant learning with a flow-matching decoder conditioned on these learned representations. We jointly train representation learning and generation in a complementary manner, where encoder features guide decoding and decoder feedback refines encoding. Consequently, UNISE preserves the original speech content while generating high-quality audio, using speech representations that are robust to noise and well-suited for effective enhancement. This leads to competitive performance on DNSMOS and speaker similarity compared to baselines in the DNS Challenge (Reddy et al., 2021) benchmark, while outperforming them on word error rate (WER) across subsets of VoiceBank-Demand (Veaux et al., 2013) and EARS (Richter et al., 2024).

## 2 METHOD

Figure 1 provides an overview of UNISE, which integrates self-supervised learning (SSL) for representation learning jointly with generative modeling. The model consists of an encoder that produces noise-invariant speech representations and a decoder that generates surface feature (e.g., variational autoencoder latent), both optimized concurrently within a single, unified framework. While Liu et al. (2025) employs masked-prediction learning for discriminative representation, we adopt the clustering approach of Chang & Glass (2024), which builds on the SwAV framework (Caron et al., 2020), to extract the essential noise-invariant representation. This method promotes invariance by clustering features and matching cluster assignments from different augmented views of the same data. Specifically, SwAV proposes swapped prediction, where the code of one view is predicted from the representation of another view. This yields a compact bottleneck that promotes the model to disentangle noise from the underlying speech content, providing a more reliable basis for generating clean speech. From these noise-invariant representations from noisy speech, the decoder reconstructs clean speech, ensuring both high-quality audio and faithful content preservation. Consequently, UNISE performs SE and learns well-designed noise-invariant speech representations for SE within a single training framework.

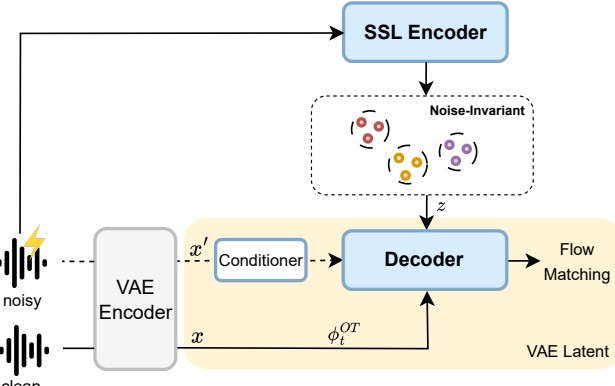

Figure 1: An overview of UNISE. The encoder is trained with noise-invariant learning, while the decoder is trained via flow matching on the VAE latent, conditioned on the representation $z$, and on randomly augmented noisy VAE latents $x'$. Both the SSL encoder and decoder are jointly optimized.

## 2.1 ROBUST SPEECH REPRESENTATION ENCODER

**Noise-Invariant Clustering**    We adopt an encoder that learns noise-invariant speech representations for speech enhancement (Figure 2) and train it jointly with the decoder. Noise-invariant training is performed by applying a clustering loss between differently distorted versions of the same input, following Chang & Glass (2024). This approach is based on Swapping Assignments between Views (SwAV) (Caron et al., 2020). Each clean waveform in a mini-batch is randomly distorted into two views by different degradation (e.g., additive noise) and fed into a transformer encoder initialized with a pretrained representation model. The output representations $z_a$ and $z_b$ are passed through a prediction head $f_\phi$, which performs linear projection and normalization. The resulting features are compared with a learnable codebook $C = [c_1, \ldots, c_V]^\top$ of $V$ codewords via scaled cosine similarities, and softmax is applied to produce the probability distributions $p_a$ and $p_b$. In practice, the target distributions $q_a$ and $q_b$ are smoothed by solving the entropy-regularized optimal transport problem to enforce full codebook usage (see Appendix A.1 for further details). Finally, we set up a swapped prediction task, minimizing the cross-entropy between the distributions from one view and the softmax probabilities of the other:

$$L_{\text{SwAV}} = l(z_a, q_b) + l(z_b, q_a), \tag{1}$$

where

$$l(z_a, q_b) = -\sum_v q_b^{(v)} \log p_a^{(v)}, \quad p_a^{(v)} = \frac{\exp(f_\phi(z_a)^T c_v / \tau)}{\sum_{v'} \exp(f_\phi(z_a)^T c_{v'} / \tau)}, \tag{2}$$

and $\tau$ is a temperature parameter. We perform online clustering on the top encoder layer. To ensure that the entire encoder, from lower to higher layers, becomes noise-robust, we fine-tune all layers.

**Auxiliary Pseudo-label Prediction**    Furthermore, unlike Chang & Glass (2024), which relies on an auxiliary task, such as acoustic piece prediction (Ren et al., 2022), to stabilize training, our framework achieves inherent stability without requiring any additional auxiliary loss. This stability is provided by the strong guidance from the decoder's reconstruction loss from Section 2.2. Nevertheless, to further enhance performance and content preservation, we also incorporate an auxiliary pseudo-label prediction loss using Sylber (Cho et al., 2025). Sylber is a sparse syllabic embedding method with strong segmentation performance, producing clusters that serve as meaningful linguistic targets, since syllables are compositional units that efficiently structure human speech perception and production. We selected syllabic features as they are directly related to speech and have proven their ability on generative tasks. Furthermore, we use this compact representation to provide a sparse and targeted guidance signal, encouraging the encoder to learn from the self-supervised objective without imposing overly strict constraints. Specifically, we generate pseudo-labels by applying k-means clustering to Sylber features extracted from clean speech, encouraging the model to learn more semantically coherent content representations. These discrete pseudo-labels are then used as

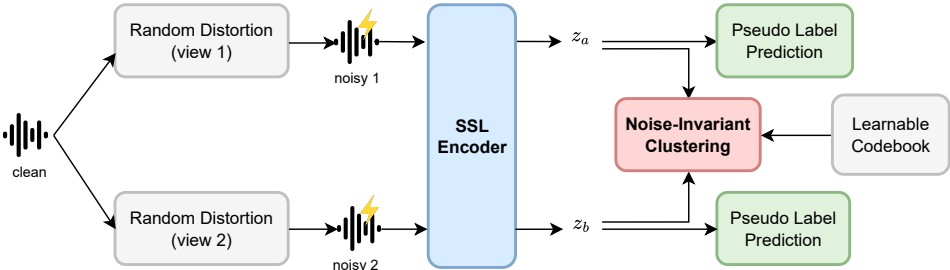

Figure 2: Illustration of the SSL encoder architecture. Two augmented views of the same clean speech are passed through a shared encoder. The model is trained with a multi-task loss that combines a clustering loss ($L_{\text{SwAV}}$) to learn noise-invariant features and an auxiliary loss ($L_{\text{Aux}}$) to predict acoustic pseudo-labels. The resulting representation, $z$, is then fed into the decoder. Refer to the main text for the definitions of the loss terms.

targets for a prediction head at the top layer, optimized with a cross-entropy loss $L_{\text{Aux}}$. The final encoder loss is:

$$L_{\text{encoder}} = L_{\text{SwAV}} + \lambda_{\text{Aux}} L_{\text{Aux}}, \tag{3}$$

where $\lambda_{\text{Aux}}$ is a scalar weight.

## 2.2 GENERATIVE SPEECH ENHANCEMENT DECODER

We train our decoder with Flow Matching (Lipman et al., 2023), a method that has proven highly effective and applicable to a wide range of speech generation tasks (Le et al., 2023; Liu et al., 2024; 2025). Flow Matching addresses the problem of generating samples $x_1$ at time $t = 1$ from a target data distribution $q(x)$ by learning a path from a prior distribution $p_0$ at $t = 0$ into a target distribution $p_1 \approx q$. In our framework, the DAC (Kumar et al., 2023) style variational autoencoder latent serves as the target distribution for speech enhancement. The flow of $x$ along this path is defined via an ordinary differential equation (ODE):

$$\frac{d}{dt}\phi_t(x) = v_t(\phi(t)), \quad \phi_0(x) = x, \tag{4}$$

where $v_t$ is a time-dependent vector field. However, the path is generally intractable because the target vector field $u_t$ is unknown. Lipman et al. (2023) consider $p_t(x|x_1) = \mathcal{N}(x|\mu_t(x_1), \sigma_t(x_1)^2 I)$ and set the conditional probability paths converge to the standard Gaussian noise distribution at $t = 0$. They propose the conditional optimal transport (OT) path by assuming $\mu_t(x) = tx_1$ and the standard deviation $\sigma_t(x) = 1 - (1 - \sigma_{min})t$, with a sufficiently small $\sigma_{min}$. Then the path is generated by the vector field $u_t(x|x_1) = \frac{x_1 - (1 - \sigma_{min})x}{1 - (1 - \sigma_{min})t}$, which is defined for $t \in [0, 1]$. This results

$$\phi_t^{OT}(x) = (1 - (1 - \sigma_{min})t)x + tx_1. \tag{5}$$

The corresponding Conditional Flow Matching (CFM) objective

$$L_{\text{CFM}} = \mathbb{E}_{t, q(x_1), p_0(x_0)} ||v_t(\phi_t^{OT}(x_0); \theta_{\text{decoder}}) - (x_1 - (1 - \sigma_{min})x_0)||^2, \tag{6}$$

where $t$ is sampled uniformly from $[0, 1]$.

In the SE framework, the model learns to generate the clean autoencoder latent $x$, typically conditioned on acoustic features and, optionally, semantic features in some works (Yang et al., 2024; Guimarães et al., 2025). We feed the decoder with the noisy autoencoder latent $x'$ as the acoustic feature, along with the encoder feature $z$ as noise-robust semantic speech representation, guiding it to reconstruct clean speech while preserving content. Following Yang et al. (2024); Guimarães et al. (2025), we employ a Conformer-based feedforward network that takes the noisy autoencoder latent as input and jointly trains the conditioner network alongside the rest of the model, enhancing latent of the input speech. The decoder incorporates the representation by adding a weighted sum of features $z$ from all encoder layers. This yields the decoder loss:

$$L_{\text{decoder}} = \mathbb{E}_{t, q(x_1), p_0(x_0)} ||v_t(\phi_t^{OT}(x_0), z, x'; \theta_{\text{decoder}}) - (x_1 - (1 - \sigma_{min})x_0)||^2, \tag{7}$$

with random dropout $x'$ with a certain probability. To prevent the decoder from depending solely on the acoustic features, which are powerful for generation but carry less semantic information, we randomly dropout the noisy acoustic feature $x'$ with a certain probability, following a classifier-free guidance strategy (Ho & Salimans, 2021).

## 2.3 JOINT TRAINING AND DOWNSTREAM APPLICATIONS

The final loss combines the encoder and decoder objectives, with both trained jointly:

$$\mathcal{L}_{\text{full}} = \mathcal{L}_{\text{encoder}} + \lambda_{\text{decoder}}\mathcal{L}_{\text{decoder}}, \tag{8}$$

where $\lambda_{\text{decoder}}$ is the decoder loss weight. While the encoder bottlenecks the essential noise-invariant information, it is simultaneously trained to reconstruct the clean target, providing suitable guidance for speech enhancement. The decoder then generates enhanced speech conditioned on this representation, along with the acoustic features. This setup enables the encoder to focus on learning information that complements acoustic features, ensuring both robust content preservation and high-quality waveform generation. All modules are trained jointly, providing feedback to each other throughout the learning process. After pre-training, the model already functions as a speech enhancement system and can be further fine-tuned on standard speech enhancement datasets for improved performance. Also, the encoder alone can serve as a robust speech representation model, which we evaluate on tasks such as robust speech recognition.

## 3 EXPERIMENTS AND RESULTS

### 3.1 PRE-TRAINING

**Datasets** Following Hsu et al. (2021); Chang & Glass (2024), we use 960 hours of speech from the LibriSpeech corpus (Panayotov et al., 2015) for pre-training. The noise datasets include DNS Challenge (Reddy et al., 2021) and SFS-Static-Dataset (Chen et al., 2021). To simulate reverberation, room impulse responses (RIRs) are randomly selected from OpenSLR28 (Ko et al., 2017), MIT IR Survey (Traer & McDermott, 2016), and EchoThief [2]. All training data are generated on the fly, with an 80% probability of adding noise at a signal-to-noise ratio (SNR) between -5 dB and 20 dB, and a 50% probability of convolving the speech with RIRs.

**Architecture Details** We pre-train a variational autoencoder for both conditioning and target distributions, following the DAC neural audio codec (Kumar et al., 2023), but replacing the quantization layers with a variational bottleneck, using publicly available speech data. We extract continuous latent features from the frozen variational autoencoder, resulting in 64-dimensional vectors at a 50 Hz frame rate. They are then normalized to zero mean and unit variance over the training set. For enhancement tasks, the features sampled by the decoder can be converted back to waveforms using the autoencoder decoder. The encoder follows the WavLM (Chen et al., 2022) architecture, consisting of 12 transformer layers with a hidden dimension of 768. We initialize from the pre-trained WavLM-Base checkpoint and fine-tune all layers during training. The conditioner branch is implemented as an 8-layer Conformer with a feed-forward dimension of 512. The decoder adopts the same architecture as UniWav (Liu et al., 2025), with the same number of layers and hidden dimensions as the encoder, a skip connection between decoder layers, and uses the ALiBi self-attention bias (Press et al., 2022). Online clustering is applied using a codebook of $V = 2048$ codewords. Sylber pseudo-label tokens are obtained by applying k-means clustering with 20k clusters to the segment-averaged Sylber[3] features extracted from the same training set used for our model. Further details on noise-invariant clustering and Sylber pseudo-label tokens are provided in Appendix A.

**Optimization** Pre-training is conducted on a NVIDIA H200 GPU using bf16 precision and gradient clipping with a threshold of 1.0. Each view contains 300 seconds of speech, which results in doubling the batch size for the decoder. We use the AdamW (Loshchilov & Hutter, 2019) optimizer with a cosine learning rate schedule that peaks at $2 \times 10^{-4}$, including 10k warmup steps, for a total of 200k updates. The full pre-training takes approximately 2.5 days. During training, the noisy VAE

---

[2] https://www.echothief.com/
[3] https://github.com/Berkeley-Speech-Group/sylber

Table 1: Comparison of various speech enhancement systems on the DNS Challenge test sets across different conditions. D and G in the 'Type' column denote discriminative and generative models, respectively. Higher DNSMOS and Spk Sim scores are better. We highlight the best results in bold and the second-best results in underline.

| System | Type | With Reverb | | | | Without Reverb | | | | Real Recordings | | |
| | | DNSMOS ↑ | | | Spk Sim ↑ | DNSMOS ↑ | | | Spk Sim ↑ | DNSMOS ↑ | | |
| | | SIG | BAK | OVL | | SIG | BAK | OVL | | SIG | BAK | OVL |
| Unprocessed | - | 1.760 | 1.497 | 1.392 | 0.941 | 3.392 | 2.618 | 2.483 | 0.969 | 3.053 | 2.509 | 2.255 |
| DEMUCS | D | 2.856 | 3.897 | 2.553 | 0.762 | 3.575 | 4.153 | 3.345 | 0.956 | 3.263 | 4.027 | 2.988 |
| FRCRN | D | 2.934 | 2.924 | 2.279 | **0.935** | 3.578 | 4.133 | 3.335 | **0.970** | 3.370 | 3.977 | 3.037 |
| CDiffuSE | G | 2.541 | 2.300 | 2.190 | - | 3.294 | 3.641 | 3.047 | - | 3.201 | 3.104 | 2.781 |
| SGMSE | G | 2.730 | 2.741 | 2.430 | - | 3.501 | 3.710 | 3.137 | - | 3.297 | 2.894 | 2.793 |
| StoRM | G | 2.947 | 3.141 | 2.516 | - | 3.514 | 3.941 | 3.205 | - | 3.410 | 3.379 | 2.940 |
| FlowSE | G | **3.601** | 4.102 | **3.331** | 0.801 | **3.685** | **4.201** | **3.445** | 0.934 | **3.635** | 4.080 | **3.263** |
| SELM | G | 3.160 | 3.577 | 2.695 | - | 3.508 | 4.096 | 3.258 | - | 3.591 | 3.435 | 3.124 |
| MaskSR | G | 3.531 | 4.065 | 3.253 | 0.827 | 3.586 | 4.116 | 3.339 | 0.929 | 3.430 | 4.025 | 3.136 |
| **UNISE** | G | 3.329 | **4.117** | 3.012 | 0.814 | 3.630 | 4.178 | 3.404 | 0.924 | 3.429 | **4.128** | 3.163 |

latents are randomly dropped with a probability of 20%. The auxiliary loss weight $\lambda_{\text{Aux}}$ is set to 0.1, and the decoder loss weight $\lambda_{\text{decoder}}$ is set to 0.25. Training samples are randomly cropped to a maximum duration of 5 seconds.

## 3.2 SPEECH ENHANCEMENT

**Experimental Setup** We evaluate UNISE's speech enhancement capability. While the model already demonstrates strong performance with pre-training alone, its enhancement quality can be further improved through fine-tuning on simulated training data generated from cleaner speech corpora. We use LibriTTS-R (Koizumi et al., 2023), VoiceBank (Veaux et al., 2013), and the deep noise suppression (DNS) challenge datasets (Reddy et al., 2021) for the clean speech dataset. And we use the same noise and RIR datasets with pre-train setup (Section 3.1). All training mixtures are generated on the fly, with an 80% probability of adding noise at a signal-to-noise ratio (SNR) randomly chosen between -5 dB and 20 dB, and a 50% probability of applying reverberation via RIR convolution. The entire model is continually trained for 30k steps on pre-trained weights, always using noisy VAE latents as input, with a learning rate of $1 \times 10^{-5}$ and the encoder frozen. To solve the initial value problems involved in generation, we employ the Euler method from `torchdiffeq` (Chen, 2018) with a step size of 0.0625. We use the test set from the publicly available DNS Challenge (Reddy et al., 2021) to compare our model speech enhancement performance with existing state-of-the-art baseline systems. We also evaluate on shared evaluation sets uploaded from DiTSE (Guimarães et al., 2025)[4], which are subsets of VoiceBank-DEMAND (VBD) (Valentini-Botinhao et al., 2016) and EARS (Richter et al., 2024) datasets.

**Baselines** We compare UNISE with several baselines. Predictive (discriminative) models include DEMUCS (Défossez et al., 2020), FRCRN (Zhao et al., 2022), and HiFi-GAN-2 (Su et al., 2021). We also classify HiFi-GAN-2 in this category, as its generator learns a deterministic mapping, although it employs adversarial training. Generative models that sample from a learned distribution include diffusion-based approaches like CDiffuSE (Lu et al., 2022), SGMSE (Richter et al., 2023), StoRM (Lemercier et al., 2023), and FlowSE (Wang et al., 2025). Other models leverage language modeling on discrete tokens include SELM (Wang et al., 2024), MaskSR (Li et al., 2024), and Genhancer (Yang et al., 2024). Finally, we include recent works that combine SSL features with flow matching or diffusion models, specifically DiTSE (Guimarães et al., 2025). DNS Challenge results are included in Table 1 where available, and metrics are newly computed on the uploaded evaluation samples (VBD and EARS), as reported in Table 2.

**Evaluation Metrics** We use DNSMOS (Reddy et al., 2022) to evaluate acoustic quality and compute speaker embedding cosine similarity following the DNS-Challenge protocol. To assess intelli-

---
[4] http://hguimaraes.me/DiTSE/

Table 2: Comparison of speech enhancement systems on the VBD and EARS test sets. Higher DNSMOS and Spk Sim scores are better, while lower WER is better. D and G in the 'Type' column denote discriminative and generative models, respectively. Best results within each type are bolded.

| System | Type | VBD | | | | | EARS | | | | |
| | | DNSMOS ↑ | | | Spk Sim ↑ | WER ↓ | DNSMOS ↑ | | | Spk Sim ↑ | WER ↓ |
| | | SIG | BAK | OVL | | | SIG | BAK | OVL | | |
| Unprocessed | | 2.953 | 2.304 | 2.138 | 0.920 | 0.0226 | 2.195 | 2.064 | 1.742 | 0.898 | 0.1008 |
| Clean | | 3.466 | 3.926 | 3.134 | - | - | 3.208 | 3.420 | 2.790 | - | - |
| VAE Reconst. | | 3.422 | 3.899 | 3.077 | 0.963 | 0.0000 | 3.202 | 3.442 | 2.789 | 0.962 | 0.0161 |
| DEMUCS | D | **3.444** | 4.001 | 3.146 | **0.895** | **0.0241** | 2.871 | 3.797 | 2.586 | 0.770 | 0.3584 |
| HiFi-GAN-2 | D | 3.402 | **4.061** | **3.150** | 0.860 | 0.0407 | **3.206** | **3.877** | **2.946** | 0.803 | 0.1958 |
| FRCRN | D | 3.384 | 4.059 | 3.096 | 0.892 | 0.0317 | 3.099 | 3.788 | 2.748 | **0.860** | **0.0894** |
| SGMSE | G | 3.387 | 4.076 | 3.140 | 0.870 | 0.0875 | 3.161 | 3.871 | 2.885 | **0.831** | 0.2842 |
| StoRM | G | 3.356 | 4.030 | 3.090 | **0.872** | 0.0799 | 3.027 | 3.648 | 2.729 | 0.797 | 0.3194 |
| Genhancer | G | 3.467 | **4.151** | **3.251** | 0.645 | 0.0543 | 3.333 | 4.016 | 3.099 | 0.697 | 0.2262 |
| DiTSE | G | **3.469** | 4.131 | 3.247 | 0.696 | 0.0407 | **3.391** | **4.035** | **3.147** | 0.753 | 0.2357 |
| **UNISE** | G | 3.424 | 4.081 | 3.163 | 0.852 | **0.0241** | 3.298 | 3.994 | 3.000 | 0.802 | **0.2063** |
| unfreeze encoder | | 3.423 | 4.104 | 3.173 | 0.832 | 0.0317 | 3.285 | 3.995 | 2.979 | 0.802 | **0.2063** |
| w/o fine-tuning | | 3.422 | 3.931 | 3.094 | 0.864 | 0.0362 | 3.232 | 3.756 | 2.863 | 0.811 | 0.2338 |

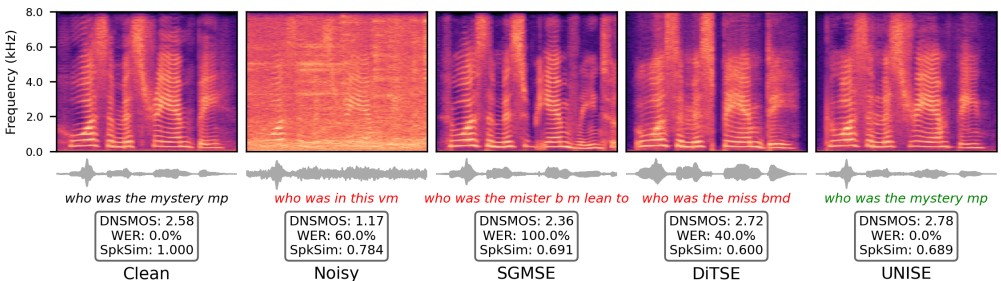

Figure 3: An example from the VBD dataset illustrates the differences among models. SGMSE produces severe hallucinations at the end of the utterance. DiTSE alters the speech structure in the middle, resulting in word substitutions. In contrast, UNISE preserves the original content.

gibility, we further report word error rate (WER) in Table 2. The official DNSMOS model from the DNS Challenge is used to assess the quality of the enhanced signals. We use WeSpeaker[5] (Wang et al., 2023) to compute speaker similarity between the enhanced and clean signals. A pre-trained Whisper-large-v3[6] model is used to transcribe the generated speech and compare it against the transcript obtained from clean speech. Both transcripts are normalized before comparison.

**Experimental Results** We first evaluate UNISE on perceptual quality (Table 1). As shown in Table 1, UNISE outperforms all other discriminative and generative baselines, with the exception of FlowSE. Even then, UNISE achieves competitive results on perceptual metrics compared to FlowSE, despite the latter having a significantly larger decoder (22 layers with 1024 channels). As with other generative models, the regression-based systems still achieve the best speaker similarity, since generative models inherently introduce some stochasticity.

Most importantly, as shown in Table 2, UNISE achieves the best WER on the VBD dataset, demonstrating strong content preservation. On the more challenging EARS dataset, UNISE achieves the best WER among generative models. While some deterministic models achieve better results, this reflects the inherent difficulty generative models face in preserving linguistic details under highly diverse and noisy conditions. However, the generative models (including UNISE) achieve superior perceptual quality (DNSMOS) compared to the discriminative baselines, supporting the performance advantage of generative approaches in fidelity. Even so, UNISE outperforms all other generative baselines, indicating that its approach helps maintain linguistic content in such challenging

[5]https://github.com/wenet-e2e/wespeaker
[6]https://github.com/openai/whisper

settings. We emphasize that Table 2 provides a system-level comparison where models may differ in size and training strategy. For a controlled evaluation isolating the contribution of the UNISE representation, please refer to the ablation study in Table 5.

Underlying these results is an inherent trade-off between perceptual quality and linguistic accuracy in generative SE models. Unlike predictive models that learn a deterministic mapping, generative models generate speech signals by sampling from a learned probability distribution. This stochastic process can lead to high-fidelity audio but may also introduce subtle phonetic shifts that impact WER. To address this, recent works (Yang et al., 2024; Guimarães et al., 2025) attempt to guide generation using pre-trained semantic features. However, these features are often not noise-robust and are not explicitly designed for speech enhancement, still resulting in suboptimal content preservation. This challenge is visibly illustrated in Figure 3: purely generative approaches like SGMSE produce hallucinations, while SSL feature guided models like DiTSE still generate incorrect words. In contrast, by leveraging noise-invariant semantic features to guide generation, UNISE achieves accurate content preservation without sacrificing the high-fidelity benefits of the generative approach. A more detailed discussion on this challenge is provided in Appendix B.1.

While pre-training alone already yields lower WER than the baselines, further fine-tuning on cleaner speech corpora substantially improves DNSMOS and reduces WER. We found that freezing the encoder during SE fine-tuning is particularly beneficial, especially for WER, because updating the encoder using only the flow-matching (generation) loss can cause the model to lose content.

## 3.3 ROBUST SPEECH RECOGNITION

We evaluate our pre-trained representation on another downstream task, robust speech recognition, to assess its performance under noisy conditions. Specifically, we test on the 1-channel track of the CHiME-4 challenge (Du et al., 2016) using the SUPERB style setup (wen Yang et al., 2021), where the encoder is frozen and only a lightweight prediction head is fine-tuned. We compare our learned representations with common self-supervised features, including HuBERT (Hsu et al., 2021) and WavLM (Chen et al., 2022). While HuBERT-MGR (Huang et al., 2021), Robust Data2Vec (Zhu et al., 2023), and R-Spin (Chang & Glass, 2024) explicitly consider distorted or noisy speech and achieve improved performance under such conditions.

**Experimental Results** See Table 3 for the results. Our encoder has a similar framework to R-Spin. However, it has an additional burden of performing reconstruction to leverage its generative capability. Despite this, its ASR performance is only slightly worse than R-Spin, which is consistent with the findings reported in UniWav (Liu et al., 2025). This gap can be partially attributed to our use of a Base-sized representation model, suggesting that larger model scales may offer further improvements. Nevertheless, our approach still substantially outperforms common SSL speech representations (Hsu et al., 2021; Baevski et al., 2020) and remains competitive with noise-robust baselines (Hsu et al., 2021; Zhu et al., 2023; Chang & Glass, 2024).

Table 3: Noisy ASR results on the CHiME-4.

| Method | WER↓ | |
| --- | --- | --- |
| | Real | Sim |
| HuBERT | 72.7 | 63.1 |
| WavLM | 52.4 | 46.4 |
| HuBERT-MGR | 49.7 | 44.3 |
| Robust data2vec | 17.5 | 20.1 |
| R-Spin | **26.4** | **26.6** |
| UNISE | 27.1 | 28.1 |

## 3.4 ABLATION STUDY

To systematically validate our framework, we conduct a comprehensive two-part study. First, we ablate the specific design choices within our pre-training recipe to identify the optimal method. Second, we evaluate the effectiveness of our learned representation on the downstream speech enhancement task. We compare our final model against two key classes of baselines: traditional generative models without semantic guidance, and systems that leverage other standard pre-trained representations. To ensure a fair comparison that isolates the impact of the representation, all systems use the same decoder architecture and are trained under identical conditions.

**Ablation on the Pre-training Method** The results, shown in Table 4, compare our final proposed method against several ablated variations. For this ablation study, we created a simulated noisy evaluation dataset by mixing speech samples from the LibriSpeech test-clean set with noise from the WHAM! dataset. dataset (Wichern et al., 2019) at random SNRs of $[-5, 0, 5, 10]$ dB. First, when

using a smaller codebook size ($V = 256$) for noise-invariant learning, the encoder tended to overfit because of insufficient codebook utilization. To mitigate this, we applied stronger regularization to promote more uniform codeword usage and prevent collapse. Nevertheless, performance degraded compared to the $V = 2048$ setting, as the larger codebook provides more fine-grained phoneme representations. Removing the random dropout applied to the noisy autoencoder latent during pre-training also degraded performance. Without dropout, the decoder can rely solely on acoustic features, which reduces the learning signal for the encoder. Dropout, in contrast, strengthens the encoder by requiring it to capture sufficient information for reconstruction even in the complete absence of acoustic features. Removing the auxiliary loss also degrades performance, highlighting its importance for robust, stable pre-training. This trend is consistent with Chang & Glass (2024), where removing auxiliary losses caused the largest performance drop, though the effect is much less pronounced in our model thanks to the decoder branch and the generation loss. Finally, applying a multi-layer clustering strategy for noise-invariant learning, as in Liu et al. (2023; 2025), which applies clustering across the top 8 layers, UNISE does not benefit from this strategy, leading instead to degraded WER performance. Note that the DNSMOS and speaker similarity metrics remained largely unchanged across these settings. In contrast, only WER degraded significantly, indicating that the encoder-related strategies primarily affect semantic aspects, with a relatively minor impact on acoustic aspects, consistent with the goal of leveraging semantic features.

Table 4: Ablation study on the pre-training method. Results are reported for the jointly trained speech enhancement model after 200k pre-training steps.

| Pre-training Configuration | DNSMOS ↑ | | | Spk Sim ↑ | WER ↓ |
|---|---|---|---|---|---|
| | SIG | BAK | OVL | | |
| UNISE | **3.601** | 4.045 | **3.312** | **0.861** | **0.0761** |
| w/o dropout AE latent | 3.577 | 4.036 | 3.284 | 0.859 | 0.0887 |
| small codebook size (V=256) | 3.568 | 4.001 | 3.257 | 0.858 | 0.0902 |
| w/o $L_{Aux}$ | 3.586 | 4.041 | 3.293 | 0.860 | 0.0914 |
| w Multi-layer Clustering | 3.598 | **4.046** | 3.308 | 0.856 | 0.1021 |

**Effect of Representations on Speech Enhancement**   For the ablation study, we freeze the encoder and train a new decoder on the speech enhancement dataset. In this setup, no dropout is applied, consistent with common SE models, and evaluation is performed on the subset of Voicebank-Demand. Table 5 compares systems built with the same decoder architecture to isolate the impact of different representations. UNISE outperforms a system that uses a frozen WavLM representation, as in previous SE models leveraging SSL features, demonstrating the effectiveness of our noise-invariant pre-training strategy. Under a controlled comparison with models of the same size, perceptual metrics are similar, but WER is significantly improved. The model without semantic representation, essentially a conventional generative speech enhancement model, results in significantly worse performance, particularly in WER. This performance gap proves the necessity of semantic features for content preservation. Further analysis is presented in Appendix B.5 and Appendix C.

Table 5: Ablation study on different encoder configurations. We train the decoder for speech enhancement models using either fixed WavLM or UNISE representations for 200k steps from scratch.

| Representation | DNSMOS ↑ | | | Spk Sim ↑ | WER ↓ |
|---|---|---|---|---|---|
| | SIG | BAK | OVL | | |
| **UNISE** | **3.290** | **4.066** | **3.032** | 0.776 | **0.0543** |
| WavLM | 3.233 | 3.970 | 2.944 | **0.781** | 0.1026 |
| w/o SSL feature | 3.174 | 3.934 | 2.873 | 0.764 | 0.1750 |

## 4  RELATED WORKS

**Generative Speech Enhancement**   Predictive SE models often suffer from over-smoothing and limited generalization. To address this, generative approaches including conditional diffusion (Lu et al., 2022; Richter et al., 2023), and Schrödinger bridge frameworks (Jukić et al., 2024; Han et al., 2025) have been explored. While improving robustness and perceptual quality, these methods often

introduce hallucination artifacts that compromise intelligibility, leading to degraded metrics such as WER and speaker similarity (Saijo et al., 2025).

To mitigate this, there have been attempts to leverage rich representation for SE (Huang et al., 2022; Hung et al., 2022). More recently, some works have incorporate pre-trained speech representations into powerful generative SE models to provide semantic guidance. For example, Yang et al. (2024); Guimarães et al. (2025) integrate continuous SSL features into generative enhancement models to provide semantic guidance. On the other hand, Wang et al. (2024); Yao et al. (2025) discretize the representations and employ language modeling approaches to capture contextual information. However, these representations are primarily designed for discriminative tasks on clean speech, limiting both their generative capability and robustness under noisy conditions.

**Robust Speech Representation**   Self-supervised learning models such as wav2vec 2.0 (Baevski et al., 2020), HuBERT (Hsu et al., 2021), data2vec (Baevski et al., 2022), and DinoSR (Liu et al., 2023) have become foundation models for discriminative tasks such as ASR or speaker identification. However, these models are primarily trained on clean speech and tend to degrade under noisy or distorted conditions. To address this, methods like WavLM (Chen et al., 2022) and others (Zhu et al., 2023; Ng et al., 2023) incorporate noisy augmentations during pre-training to enhance robustness. Chang & Glass (2024) proposed R-Spin, which encourages noise-robust representations by integrating noise-invariant clustering under noisy conditions, a method we adopt in this work. However, simply applying noise augmentation does not guarantee noise-invariant representations; the resulting features may remain highly correlated with non-speech sounds Gong et al. (2023), limiting generalization. While these approaches improve robustness, they primarily target discriminative tasks and still lack strong generative capability.

**Pre-training for Generative models in Speech**   While pre-training in speech has traditionally focused on discriminative tasks, recent works have explored generative pre-training. VALL-E (Chen et al., 2025) and Voicebox (Le et al., 2023) demonstrated strong zero-shot adaptation by training large-scale generative models conditioned on text. SpeechFlow (Liu et al., 2024) further introduced a generative pre-training approach using untranscribed speech with Flow Matching and masked conditions. Building on these ideas, UniWav (Liu et al., 2025) proposed a unified model jointly trained for both flexible representation learning and generation. UNISE extends these improvements to the speech enhancement task by conditioning the generative model on noisy inputs rather than masked ones, and on noise-invariant representations rather than text or clean representations, enabling it to operate effectively under realistic distortions.

## 5 CONCLUSION

In this work, we introduce a unified framework for SE that combines a self-supervised encoder and a generative decoder. The encoder is trained to produce robust, noise-invariant representations, which are then used to guide the decoder's generation process. Previous SE models often leverage frozen standard SSL features, which lack robustness to noise and are not optimized for generation. In contrast, UNISE conditions the decoder on a specialized, fine-tuned bottleneck through jointly noise-invariant learning. As a result, unlike common representations, whose mutual information with phonemes decreases as SNR decreases, our representation remains consistent across SNRs (Appendix B.4), enabling effective preservation of linguistic content while removing complex distortions. By addressing major limitations of previous systems, this approach ensures robust and faithful content preservation.

**Limitations**   Since our model focuses on SE, Liu et al. (2025) also notes that there remains a gap in the discriminative task at similar scales. We also notice a slight performance drop in our model on recognition tasks. The open research problem is to improve the noise-robustness while simultaneously ensuring strong performance on both discriminative and generative tasks. We leave a thorough investigation of alternative design choices, such as distilling to a clean representation (Guimarães et al., 2023) and exploring decoder architectures that operate on discrete representations (Wang et al., 2024; Yang et al., 2024; Yao et al., 2025), as important directions for future work. Additionally, scaling training to larger, more diverse corpora may further improve the model's generalization capabilities and enable evaluation under more realistic, complex acoustic conditions.

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

# A METHOD DETAILS

## A.1 NOISE-INVARIANT CLUSTERING DETAILS

Unlike common contrastive learning methods that rely on explicit pairwise feature comparisons, the approach we use, SwAV (Caron et al., 2020), promotes invariance by clustering features and matching cluster assignments from different augmented views of the same data. They propose a swapped prediction to predict the code of a view from the representation of another view. In our case, we augment the speech with additive noise and reverberation to achieve noise invariance.

SwAV performs a swapped prediction task by minimizing the cross-entropy between the distributions of one view and the softmax probabilities of the other. The representations $z_a$ and $z_b$ are passed through a prediction head $f_\phi$, then compared with a learnable codebook $C = [c_1, \ldots, c_V]^\top$ of $V$ codewords using scaled cosine similarities, followed by softmax to produce probability distributions.

To implement this in an online fashion, the codes are computed using only the representations within a batch. The target distribution $Q = [q_1, \ldots, q_B]^\top$ is obtained by mapping representation vectors $Z = [z_1, \ldots, z_B]^\top$. It is optimized to maximize the similarity between the features and their corresponding codebook $C$:

$$Q^* \in \arg\max_Q \mathrm{Tr}(QCZ^\top) + \epsilon H(Q), \tag{9}$$

where $H(Q) = -\sum_{ij} Q_{ij} \log Q_{ij}$ is the entropy function and $\epsilon$ is a parameter that controls the smoothness of the mapping. The entropy regularization encourages the use of all codewords and prevents collapse to a single cluster, but a higher $\epsilon$ can lead to a unique representation by strong regularization.

The optimal target distribution $Q^*$ is obtained by solving the entropy-regularized optimal transport problem in Eq. (9). Following Caron et al. (2020), this can be efficiently computed using the iterative Sinkhorn–Knopp algorithm on GPUs, which alternately normalizes the rows and columns of the matrix $\exp(CZ/\epsilon)$ to satisfy the equipartition constraints. This procedure yields a soft assignment matrix $Q^* = \mathrm{Diag}(u) \exp(CZ^\top/\epsilon) \mathrm{Diag}(v)$, ensuring that each codeword is assigned to approximately the same number of samples within a batch. Following Spin (Chang et al., 2023), we set the temperature to $\tau = 0.1$ and perform 3 Sinkhorn iterations. For the codebook size $V = 2048$, we use a smoothness parameter of $\epsilon = 0.02$, while for the smaller codebook $V = 256$, we increase it to $\epsilon = 0.05$ to mitigate collapse.

## A.2 SYLBER PSEUDO-LABEL GENERATION DETAILS

Sylber pseudo-label tokens are obtained by applying k-means clustering with 20k clusters to the segment-averaged Sylber[7] features extracted from the same training set used for our model. Clustering is performed using the `MiniBatchKMeans` implementation from the `scikit-learn` (Pedregosa et al., 2011) package, with a mini-batch size of 10,000 frames and the k-means++ (Arthur & Vassilvitskii, 2006) initialization strategy with 5 random restarts. The resulting discrete cluster indices are then expanded back to match the durations of the original segments based on the frame alignments provided by the Sylber model, while non-speech regions are filled with a null token.

---

[7]`https://github.com/Berkeley-Speech-Group/sylber`

## B  ADDITIONAL EXPERIMENTS AND DISCUSSIONS

### B.1  TRADE-OFF BETWEEN CONTENT PRESERVATION AND PERCEPTUAL QUALITY IN GENERATIVE SE

While DNSMOS reflects perceptual quality and WER captures linguistic accuracy, these two aspects can sometimes conflict, particularly in generative SE models. Enhancement often improves perceptual quality, but linguistic and speaker content can be altered, especially for challenging and highly diverse samples such as those in EARS. This issue is more pronounced in generative SE models, which reconstruct the speech signals by sampling from a learned distribution. Unlike predictive (discriminative) models that directly estimate a deterministic mapping, generative models inherently allow more flexibility, which can lead to subtle shifts in phonetic details or speaker characteristics. This trend aligns with findings reported in the URGENT Challenge 2025 (Saijo et al., 2025).

To address this, recent works (Yang et al., 2024; Guimarães et al., 2025) attempt to guide generation using pre-trained semantic features. However, these features are often not noise-robust and are not explicitly designed for speech enhancement, still resulting in suboptimal content preservation. In contrast, our noise-invariant learning explicitly encourages the encoder to retain information that should be preserved regardless of noise. This representation then guides the generative decoder to reconstruct the clean speech while maintaining the linguistic content. Consequently, UNISE achieves lower WER than other generative baselines, sometimes at a small cost in perceptual metrics, DNSMOS (Table 2). This reflects a natural trade-off: prioritizing content preservation can slightly limit the decoder's ability to maximize perceptual quality. However, in our ablation study (Table 5) with the same decoder, replacing other SSL features with UNISE led to improvements in both WER and DNSMOS. This suggests that with a larger decoder or optimized training strategy, it may be possible to further improve both linguistic accuracy and perceptual quality in future work.

### B.2  SUPERB SE

To benchmark the effectiveness of the UNISE representation for speech enhancement (SE), we follow the SE downstream task in the official SUPERB recipe [8] (wen Yang et al., 2021; Tsai et al., 2022), which is based on the VoiceBank-DEMAND dataset (Valentini-Botinhao et al., 2016). Our model achieves the best scores across all metrics in this benchmark, demonstrating that the proposed representation is well-suited for SE.

Table 6: SUPERB enhancement_stft2 results.

| Representation | SI-SDR ↑ | STOI ↑ | PESQ ↑ |
|---|---|---|---|
| HuBERT | 9.2117 | 0.9487 | 2.9924 |
| WavLM | 9.4384 | 0.9487 | 2.9857 |
| **UNISE** | **9.5711** | **0.9506** | **3.0385** |

### B.3  SYLLABLE DETECTION AND DISCOVERY

Since our encoder is also trained with an auxiliary objective to predict Sylber features, the UNISE representation can additionally be applied to syllable detection and discovery. While Sylber generates a clean and robust segment structure through self-segmentation distillation alone, UNISE is jointly trained with additional objectives that are not explicitly segmented, resulting in slightly noisier encoder outputs. For segmentation, we apply the MinCut algorithm with a normalization threshold of 0.1, a merge threshold of 0.4, and an estimated syllable duration of 100 ms. As shown in Table 7, our model performs worse than models specifically trained for segmentation, but achieves better performance than HuBERT, a common representation trained purely in a self-supervised manner. We report these results to demonstrate that the UNISE representation can be effectively applied to downstream tasks such as this one, even without achieving state-of-the-art performance.

Table 7: Syllable detection and discovery results measured. Pr: precision, Re: recall, R: R-value, SP: syllabic purity, CP: cluster purity, and MI: mutual information. All metrics are measured using LibriSpeech test data.

| Model | Syllable Detection | | | | Syllable Discovery | | |
|---|---|---|---|---|---|---|---|
| | Pr↑ | Re↑ | F1↑ | R↑ | SP↑ | CP↑ | MI↑ |
| HuBERT (Hsu et al., 2021) | 51.4 | 31.4 | 39.0 | 50.1 | 33.1 | 28.4 | 3.54 |
| VGHuBERT (Peng & Harwath, 2022) | 65.3 | 64.3 | 64.8 | 70.0 | 53.4 | 43.6 | 4.66 |
| SDHuBERT (Cho et al., 2024) | 64.3 | **71.0** | 67.5 | 70.7 | 54.1 | **46.2** | 4.76 |
| Komatsu & Shinozaki (2024) | 73.3 | 67.6 | 70.3 | 74.6 | 59.4 | 44.5 | 5.08 |
| Sylber (Cho et al., 2025) | **76.6** | 68.3 | **72.2** | **75.9** | **64.0** | 43.9 | **5.28** |
| **UNISE** | 56.7 | 47.6 | 51.8 | 59.8 | 37.6 | 22.1 | 4.08 |

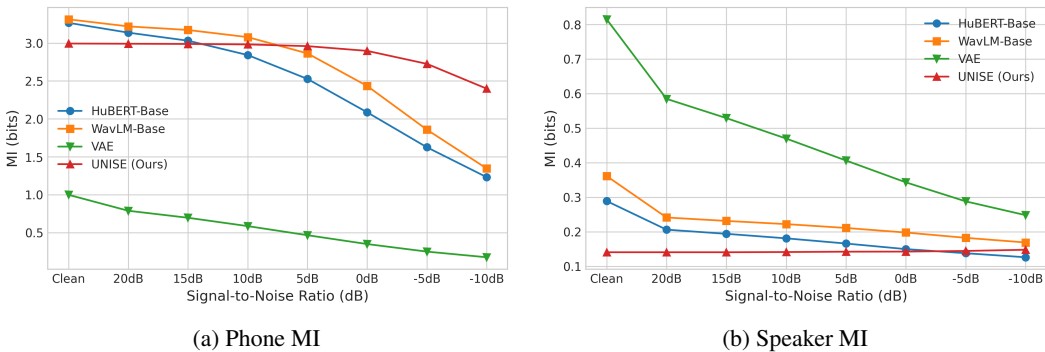

(a) Phone MI            (b) Speaker MI

Figure 4: Mutual information (MI) between speech representations from the 7th layer of HuBERT, the 11th layer of WavLM, and the final layer of UNISE and the content labels for phoneme and speaker across varying signal-to-noise ratios (SNRs).

### B.4 MUTUAL INFORMATION BETWEEN REPRESENTATION AND LABELS

To investigate our learned representations further, we measure mutual information (MI) with respect to phoneme and speaker labels, following the protocol in Hsu et al. (2021). Unlike the original setting, we also evaluate under noisy conditions. Specifically, we first perform $k$-means clustering on clean representations and map the resulting clusters to ground-truth labels to obtain the joint probability distribution, as is commonly done. We then fix this cluster-to-label mapping and assign the noisy representations to the closest clusters, computing MI based on this assignment. This procedure allows us to quantify how well the representations preserve phoneme and speaker information in the presence of noise.

In Figure 4a, the mutual information of WavLM and HuBERT degrades sharply as the SNR decreases. In contrast, UNISE is much more robust to noise and retains higher information at low SNR levels. Although our model contains slightly less phoneme-related information overall—especially on clean data—our representations exhibit stronger generative ability, as shown earlier. Moreover, the autoencoder's latent features exhibit lower mutual information with linguistic content, focusing instead on generation.

In Figure 4b, the autoencoder latent encodes a large amount of speaker information, which is closely tied to acoustic characteristics. For other representations, our model maintains more consistent mutual information with speaker labels across SNRs, similar to the phoneme case. However, since speaker information is influenced by additive noise and reverberation, our noise-invariant training reduces the amount of speaker-related information. Nevertheless, speech enhancement models are jointly trained with acoustic features such as mel-spectrograms or autoencoder latents, which in-

---

[8]github.com/s3prl/s3prl/tree/main/s3prl/downstream/enhancement_stft2

herently contain rich speaker information. As a result, these models can still achieve high speaker similarity, even if the learned representation encodes less explicit speaker information.

## B.5 DNSMOS vs PER

We analyze the relationship between perceptual quality and intelligibility across different feature conditioning for individual samples. Figure 5 presents scatter plots of phoneme error rate (PER) versus DNSMOS values on a severely noisy dataset. For this analysis, we generated simulated noisy mixtures by combining speech from the LibriSpeech (Panayotov et al., 2015) test-clean set with noises from the WHAM! (Wichern et al., 2019) dataset at an SNR of -5 dB. We compare ablated generative SE models in Table 5 using no SSL representations, WavLM features, and UNISE features.

Without semantic features, simple generative SE models also may achieve high perceptual scores while failing to preserve linguistic content. With SSL features, it improves, but models can still generate incorrect words due to limited noise robustness. In contrast, our model consistently preserves linguistic content while achieving high perceptual quality. Notably, other models may sound plausible yet be incorrect, showing high DNSMOS despite poor intelligibility. This phenomenon has also been observed in Saijo et al. (2025), which reported occasional hallucinations in generative SE models. Even when perceptual quality is relatively low, UNISE accurately preserves linguistic content. This stability demonstrates that UNISE provides both reliable perceptual enhancement and faithful content preservation, effectively avoiding hallucinations.

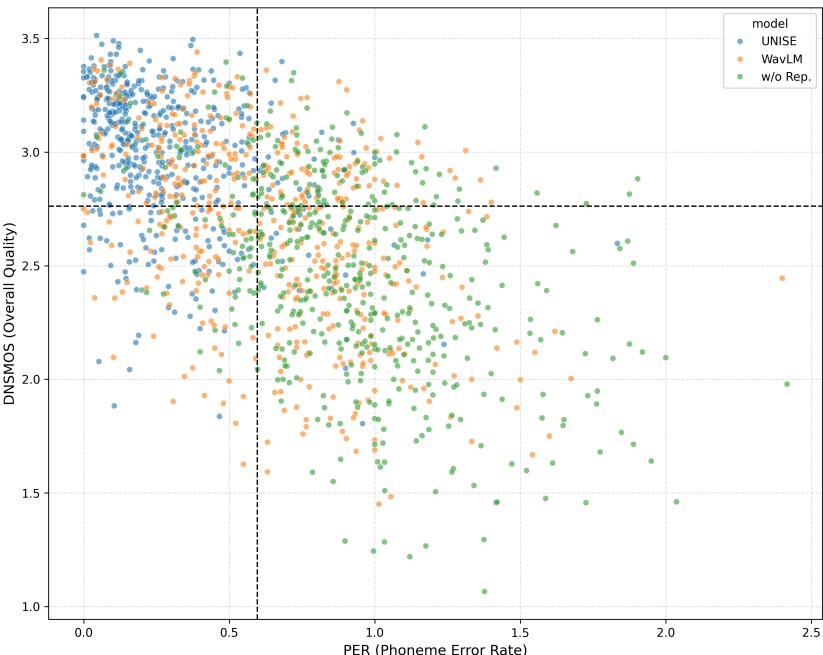

Figure 5: DNSMOS vs. PER on the -5 dB SNR dataset. Dashed lines indicate medians.

## C VISUALIZED SAMPLE

To better illustrate the hallucination behaviors discussed in the main paper, Figure 6 shows an example (referenced in Appendix B.5) comparing the ablated models introduced in Section 3.4. The model without SSL features produces large hallucinations, inserting unrelated content and breaking the original utterance structure. Using WavLM features stabilizes the enhancement, but minor word errors still appear. In contrast, UNISE preserves the content and alignment accurately, showing no observable linguistic errors. All models achieve DNSMOS scores above 3.00, which would typically suggest perceptually good quality. Nevertheless, substantial linguistic errors exist in all models

except UNISE, highlighting that DNSMOS can remain high even when semantic fidelity is poorly preserved or hallucinations occur. In contrast, our model achieves both good perceptual quality and strong content preservation.

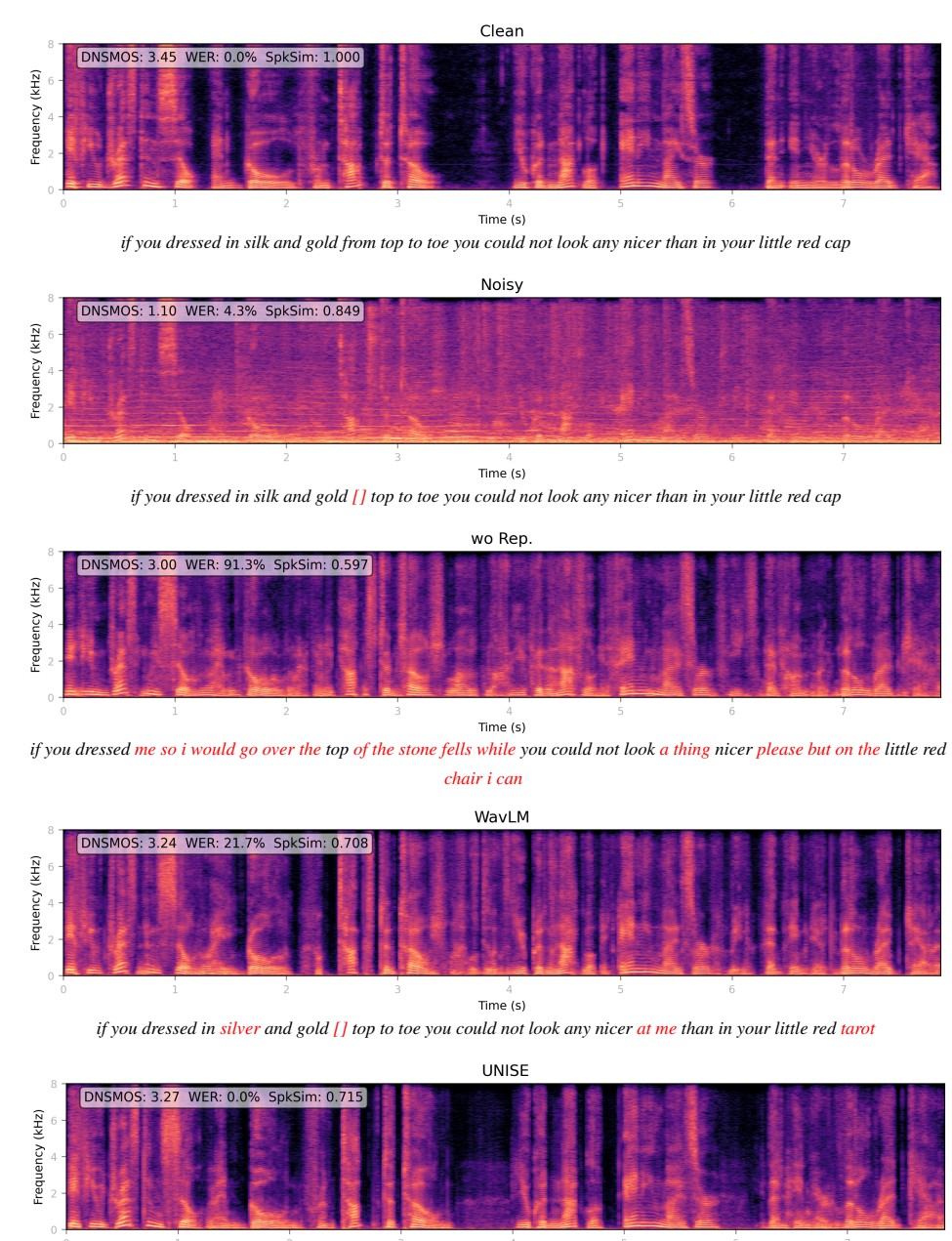

Figure 6: Example comparison of enhanced outputs across ablated models. Red text highlights incorrect words, and corresponding metric values are provided for each model.

