# OpenReview forum: "UNISE: Unified Noise-Invariant Learning for Speech Enhancement toward Improved Content Preservation"
_ICLR.cc/2026/Conference — Submitted to ICLR 2026_

### Official Review · Reviewer_rWcK · 2025-10-29

**Soundness:** 2
**Presentation:** 2
**Contribution:** 3
**Rating:** 4
**Confidence:** 4

**Summary:**

The paper introduces a "unified generative SE" model called UNISE. The model aims to incorporate a noise-invariant representation to improve robustness and content preservation. The noise-invariant representation is achieved by defining a learning target that maps multiple views of the same clean speech under different distortions to the same target cluster. The clustering objective is employed after each layer of the proposed encoder. A syllabic label prediction task is used as an auxiliary loss to improve content preservation. The noise invariant encoder is combined with a generative decoder employing flow-matching. The authors compare their model to multiple other competing models and demonstrate that their approach yields benefits in content preservation against most of the tested models. They legitimize their findings by multiple ablations or further tests on ASR performance, different loss combinations and a consideration of mutual information between their learned embedding and a phone target across multiple SNRs.

**Strengths:**

The problem of content preservation in generative models is indeed a relevant and timely research topic, which makes the proposal interesting for the community.

The proposed method follows past research and expands upon it in a plausible way. The proposed method achieves a reduction in word error rate over compared generative speech enhancement models while achieving comparable DNSMOS and SpkSim values, indicating preserved quality.

We appreciated the audio samples, which support the claim of increased content preservation in contrast to the also featured SGMSE.

Appendix 2 is a welcome additional analysis comparing the mutual information between the encoder output and corresponding phone labels, which shows a high degree of mutual information  that remains high under low SNRs in case of the proposed model.

**Weaknesses:**

While the paper is well-written w.r.t. phrasing, grammar and overall clarity, aspects of the presentation inhibit effortless understanding.

Mathematical notations and figures have alignment issues.
- Section 2.1 introduces output representations z_a and z_b but Figure 2 uses different labels z_1 and z_2 for presumably the same concept?
- Equation (3) includes a summation over k, which is not explicitly defined.
- The pseudo-label prediction task is defined as y but y is never referred anywhere. The same goes for the Sylber features s, which do not occur in any figure or equation thereafter.
- It is furthermore unclear what the index c in Equation (1) means. Defining (capital) C as a learnable codebook does not obviously define the summation.

We find that a very muc h intensified connection between the figures and their textual description is necessary. In the aforementioned cases, the missing definition of symbols made a verification of mathematical correctness impossible.

Furthermore, model sizes are not included in Table 1 and Table 2. It is therefore hard for the reader to separate architecture and model design from model size. The textual discussion references model size on multiple occasions, but no number for the parameter count is mentioned for any model.

The dataset that Table 1 reports on is only implied to be the DNS Challenge test set in Section 3.2. The table's caption should state the dataset clearly.

In Appendix A.1, you state "As shown in Table 6, our model achieves better results than common representation." but Table 6 clearly shows most models beating your model in all metrics. There must be a mistake in either the table or your argument.

The appendix is not referenced anywhere in the text. If space permits an introductary sentence in the main text helps to contextualize the appendix.

**Questions:**

The discussion of Table 2 is not sufficiently discussing the two test sets independently. It is only stated that UNISE achieves best WER for linguistic reconstruction. However it slightly falls behind HiFi-GAN-2 on the EARS test set. On EARS---with DNSMOS only slightly higher than HiFi-GAN-2 and SpkSim slightly below HiFi-GAN-2---the benefit of UNISE over HiFi-GAN-2 is not self-evident. Could you clarify your phrasing in this respect?

Also, did your analysis show a reason for the less pronounced benefit on EARS?

---

> ### Author Response · Authors · 2025-11-21
> **Response to Reviewer rWcK (1/2)**
>
> We thank Reviewer rWcK for the exceptionally careful reading of the manuscript and for identifying several critical issues in clarity, notation, and presentation. We sincerely apologize for these errors and will correct all of them in the revision.
>
> **W1, W3 -- Notation Inconsistencies and Dataset Ambiguity.**
> We thank the reviewer for carefully identifying these issues. All inconsistencies in figures, symbols, and notation have been corrected in the revision, and the notation has been updated consistently throughout the paper. In addition, the caption for Table 1 has been revised to clearly state that all comparisons were conducted on the DNS Challenge test sets under the specified conditions. Thank you again for pointing out these problems.
>
> **W2 -- Missing Model Sizes in Tables.**
> We agree that parameter counts can help separate architectural design from model capacity. However, several baseline papers do not report their exact parameter sizes, making it difficult to provide a complete and fair comparison across all models.
>
> Instead, we address this concern through a capacity controlled ablation study (Section 3.4), where the decoder architecture and training setup are fixed and only the encoder is changed. This setup isolates the contribution of our proposed representation from model capacity, even without full parameter-count information for all baselines.
> Moreover, on the SE task from the SUPERB benchmark, our representation achieves superior performance compared to other SSL features (HuBERT and WavLM) under the same setup, further supporting that the gains stem from the representation rather than model capacity. This benchmark result is included in the revision as an additional fair-capacity comparison across different representations (Appendix B.2).
>
> **W4 -- Appendix Table 6 Misstatement.**
> We apologize for the confusion. What we meant to say was that while our model performs worse than models specifically trained for segmentation, it achieves better performance than HuBERT, a common representation trained purely in a self-supervised manner. We correct this in the revision.
>
> **W5 -- Appendix.**
> Thank you for this valuable suggestion. We contextualize the appendix within the main text to improve the paper's logical flow.

---

> > ### Comment · Reviewer_rWcK · 2025-11-25
> > **Reviewer Response to Rebuttal 1/2**
> >
> > W1,W3: Thank you for addressing the issues. The alignment of notations between figures and mathematical description has improved. The notation of the loss function is also improved and contains no obvious error.
> > W2: We appreciate clarifying the additional purpose of this ablation. The ablation study targeting isolated encoder changes is indeed very useful and a good way of isolating the effect of your architectural change. Explicitly motivating the choice of separating the assessment of this ablation from the main results table (Table 2) might prevent confusion as the text mentions incomparable model sizes in different places. This solidifies Table 2’s identity largely as a system comparison that does not allow for isolating particular model qualities apart from the categorization discriminative/generative. Please see our remarks to this topic in our response to Q1,Q2.
> >
> > W4: Thank you for correcting the misstatement. The corrected statement is now correct but a far weaker statement than the initial one.

---

> > > ### Author Response · Authors · 2025-11-27
> > > **Response to Reviewer rWcK (1/2)**
> > >
> > > We sincerely thank the reviewer for the continued engagement. Reviewer's feedback has been constructive in improving the clarity and presentation of our manuscript.
> > >
> > > **W2:**
> > >  We agree with the suggestion to more explicitly motivate the separation of this controlled assessment from the main results.
> > > Accordingly, we have revised the manuscript to explicitly state that Table 2 provides a system-level comparison where models may differ in architecture and training strategy. We now clearly direct readers to the ablation study (Table 5) for a controlled evaluation that isolates the specific contribution of the UNISE representation.
> > >
> > > Furthermore, we have categorized the baseline models into discriminative and generative types and added detailed analysis regarding the performance characteristics of each category. Please see our response to Q1 and Q2 for further details on this point.
> > >
> > > **W4:**
> > >  We included the performance on syllable detection and discovery to showcase the versatility of the UNISE representation to downstream applications beyond its primary enhancement or noisy ASR. We have clarified this intent in the revised manuscript.

---

> ### Author Response · Authors · 2025-11-21
> **Response to Reviewer rWcK (2/2)**
>
> **Q1, Q2 -- Discussion of Table 2 and EARS Performance.**
> While DNSMOS reflects perceptual quality and WER captures linguistic accuracy, these two aspects can sometimes conflict, particularly in generative SE models. Enhancement often improves perceptual quality, but linguistic and speaker content can be altered, especially for challenging and highly diverse samples such as those in EARS. This issue is more pronounced in generative SE models, which reconstruct the waveform by sampling from a learned distribution. Unlike predictive models that directly estimate a deterministic mapping, generative models inherently allow more flexibility, which can lead to subtle shifts in phonetic details or speaker characteristics.
> This trend aligns with findings reported in the URGENT Challenge 2025 [1].
> To address this, our noise-invariant learning explicitly encourages the encoder to retain information that should be preserved regardless of noise. This representation then guides the generative decoder to reconstruct the clean speech while maintaining the linguistic content. Consequently, UNISE achieves lower WER than other generative baselines. It already demonstrates better performance than HiFi-GAN-2 on VoiceBank-Demand, while on EARS the gains are smaller—though UNISE still achieves the best WER among generative models. We anticipate that training with more diverse and challenging datasets could further improve performance on harsh datasets like EARS in future work.
>
> **Closing.**
> We thank the reviewer again for the detailed and constructive feedback. These corrections significantly improve the clarity and accuracy of our paper. We have updated the notation, tables, dataset captions, and result discussions in accordance with the comments. We welcome any further questions or suggestions.
>
> **References**
> - **[1]** Saijo et al., Interspeech 2025 URGENT speech enhancement challenge., Interspeech 2025.

---

> > ### Comment · Reviewer_rWcK · 2025-11-25
> > **Reviewer Response to Rebuttal 2/2**
> >
> > Q1,Q2: We agree with your remarks about generative models as found by Saijo et al.  While the different WER rank orders may be partially attributed to the harshness of this particular noisy version of the EARS dataset and the categorization discriminative/generative, the design of the table and the accompanying text do not actively to prevent this question from arising. While you earlier introduce HiFi-GAN-2 as a discriminative model, neither the table nor the text contains group labels. Your assessment may be correct but the presentation prevents the reader from achieving the same conclusions without major effort, even in the revised version.
> >
> > Closing:
> > The improved clarity allowed for an assessment of your loss function which is likely correct and without obvious errors. The additional reproducibility and implementation details as requested by other reviewers are also appreciated. We are therefore open to raising the soundness rating to 3. We suggest improving presentation further by explicitly clarifying and motivating the purposes of tables (e.g., Table 2, Table 5).

---

> > > ### Author Response · Authors · 2025-11-27
> > > **Response to Reviewer rWcK (2/2)**
> > >
> > > **Q1,Q2:**
> > >  We agree with the reviewer that the previous presentation of results hindered a clear understanding of the performance differences between discriminative and generative models. While we had included some discussion in Appendix B.1 of the previous revision, we acknowledge that it was not sufficiently linked to the main results table, making it difficult for readers to contextualize the findings without significant effort.
> > >
> > > To address this, we have revised the manuscript in the following ways:
> > >
> > > * **Explicit Categorization:** We have explicitly categorized the baseline models into "discriminative" and "generative" groups, a distinction visually marked by the Type column in Table 2. Furthermore, we refined the list of discriminative models, adding DEMUCS and FRCRN to Table 2 to provide stronger evidence for comparative analysis (Miipher was removed as its classification was not sufficiently clear for this discussion).
> > >
> > > * **Integrated Discussion:** We have added the deeper discussion regarding the trade-offs inherent in generative models (such as the WER rank orders on harsh noise) also into the main text. This analysis is now directly integrated with the interpretation of Table 2 and is further supported by qualitative examples presented in Figure 3.
> > >
> > > **Closing.** We are grateful for the reviewer's helpful comments and for considering raising the soundness rating. Following the reviewer's specific suggestion, we have explicitly clarified the distinct roles of the main results (Table 2) and ablation tables (Table 5) and integrated a deeper discussion on trade-offs in generative SE models into the main text. We believe these revisions will make it significantly easier for readers to understand the purpose and results of our work, effectively resolving the ambiguity in presentation.

---

### Official Review · Reviewer_zNt5 · 2025-10-31

**Soundness:** 2
**Presentation:** 2
**Contribution:** 2
**Rating:** 4
**Confidence:** 4

**Summary:**

This paper proposes a unified framework that combines noise-invariant self-supervised representation learning with generative speech enhancement to address limitations of existing methods. Experimental results demonstrate empirical performance outperforming baselines on WER for content preservation while maintaining competitive perceptual quality and speaker similarity.

**Strengths:**

- The paper identifies two unresolved challenges in SE using SSL representations, noise robustness and generative capability, and proposes a unified solution. This addresses a critical gap between discriminative SSL and generative SE.

- The noise-invariant contrastive learning explicitly disentangles speech content from noise, a improvement over prior noise-augmented SSL that only mitigates noise rather than isolating content.

- The joint training of encoder and flow-matching decoder creates a feedback loop: the encoder learns content-preserving features guided by the decoder’s reconstruction loss, while the decoder uses these features to avoid hallucinations.

**Weaknesses:**

- The paper states Sylber provides "sparse syllabic embeddings" but does not clarify: (i) how Sylber features are extracted from clean speech, (ii) why syllabic pseudo-labels are more effective than phonetic or word-level labels for content preservation, (iii) how non-speech frames are "filled with null tokens", what defines a non-speech frame?

- The paper emphasizes UNISE’s strength in WER but does not address a critical question: Does the focus on WER come at the cost of perceptual quality in extreme noise? For example:In Table 1, UNISE’s DNSMOS score (3.334) is lower than FlowSE’s (3.601) under non-reverberant conditions. Is this a necessary tradeoff for better WER, or can the model be adjusted to improve both?

- UNISE’s encoder has lower mutual information with speaker labels than WavLM/HuBERT, but the model still achieves competitive speaker similarity (Table 1). The paper attributes this to "acoustic features in the decoder," but this is vague.

**Questions:**

- Noise-Invariant Contrastive Learning DetailsCodebook Design: The paper mentions a learnable codebook with V=2048 codewords but provides no details on: (i) how codewords are initialized (e.g., random vs. pre-clustered on clean speech), (ii) how optimal transport "smooths target distributions", what OT cost function is used? (iii) why V=2048 was chosen (ablation over codebook size would strengthen this choice).

---

> ### Author Response · Authors · 2025-11-21
> **Response to Reviewer zNt5 (1/2)**
>
> We thank Reviewer zNt5 for the constructive evaluation and for highlighting the strengths of our unified approach. Below we respond to the raised concerns.
>
> **W1 -- Clarification of Sylber Pseudo-Labels and Non-Speech Frames.**
> We apologize for the missing detail and have clarified it in the revision (Section 3.1).
>
> Sylber pseudo-label tokens are obtained by applying k-means clustering with 20k clusters to the segment-averaged Sylber [1] features extracted from the same training set used for our model. Clustering is performed using the `MiniBatchKMeans` implementation from the `scikit-learn` package, with a mini-batch size of 10,000 frames and the k-means++ initialization strategy with 5 random restarts. The resulting discrete cluster indices are then expanded back to match the durations of the original segments based on the frame alignments provided by the Sylber model, while non-speech regions are filled with a null token.
>
> Regarding the pseudo-label design, while we could not experiment with alternative pseudo-label types, our ablation study (as shown in Table 4) did confirm that the inclusion of this auxiliary loss significantly improves content preservation, validating its role in our framework.
> We chose Sylber features because they demonstrate superior performance compared to common representations (e.g., HuBERT) in resynthesis and provide a compact, targeted pseudo-label signal as a weak auxiliary. However, investigating alternatives, such as phonetic-, word-, or syllabic-level pseudo-labels like ours, represents a valuable direction for future work.
>
> **W2 -- Tradeoff Between WER and DNSMOS.**
> This is an excellent question. While DNSMOS reflects perceptual quality and WER captures linguistic accuracy, these two aspects can sometimes conflict, particularly in generative SE models.
> Enhancement often improves perceptual quality, but linguistic and speaker content can be altered, especially for challenging and highly diverse samples such as those in EARS. This issue is more pronounced in generative SE models, which reconstruct the waveform by sampling from a learned distribution. Unlike predictive (discriminative) models that directly estimate a deterministic mapping, generative models inherently allow more flexibility, which can lead to subtle shifts in phonetic details or speaker characteristics.
> This trend aligns with findings reported in the URGENT Challenge 2025 [2].
>
> To address this, our noise-invariant learning explicitly encourages the encoder to retain information that should be preserved regardless of noise. This representation then guides the generative decoder to reconstruct the clean speech while maintaining the linguistic content.
> Consequently, UNISE achieves lower WER than other generative baselines, sometimes at a small cost in perceptual metrics, DNSMOS. This reflects a natural trade-off: prioritizing content preservation can slightly limit the decoder’s ability to maximize perceptual quality. However, in our ablation study (Section 3.4) with the same decoder, replacing other SSL features with UNISE led to improvements in both WER and DNSMOS. This suggests that with a larger decoder or optimized training strategy, it may be possible to further improve both linguistic accuracy and perceptual quality in future work.
>
> **W3 -- Speaker MI vs. Speaker Similarity.**
> The reviewer is correct that the UNISE encoder has low speaker MI, which is affect from noise-invariant learning. However, the decoder's total input consists of both the UNISE encoder representation and a VAE codec latent. And as shown in the MI figure, this VAE codec latent retains high MI with speaker labels, as it encodes the acoustic information.
>
> Therefore, the model achieves high speaker similarity by combining the feature from the UNISE encoder with the acoustic and speaker information from the noisy VAE latent directly. This result is consistent with many SE models that use a primary acoustic feature, such as a codec or spectrogram.

---

> ### Author Response · Authors · 2025-11-21
> **Response to Reviewer zNt5 (2/2)**
>
> **Q1 -- Codebook Initialization, OT Cost, and Vocabulary Size.**
> We apologize for omitting these details.
>
> * **(i) Initialization**: The codebook $C$ is initialized randomly.
> * **(ii) OT Cost**: The target distribution $Q=[q_1,...,q_B]^\top$ is obtained by mapping representation vectors $Z=[z_1,...,z_B]^\top$.
>     The target distribution $Q$ is optimized to maximize the similarity between the features and their corresponding codebook $C=[c_1, \dots ,c_V]^\top$:
>     $$
>     Q^* \in arg \max_Q \text{Tr}(QCZ^\top) + \epsilon H(Q),
>     $$
> where $C$ is the codebook, $Z$ are the feature representations, $H(Q)$ is the entropy function, and $\epsilon$ controls the smoothness of the distribution. The optimal $Q^*$ is efficiently computed using the Sinkhorn--Knopp algorithm (Appendix A).
> Please check the revision for more details.
> * **(iii) Choice of $V=2048$**: Due to computational constraints, we were unable to perform an exhaustive experimental sweep of all design choices. However, we did investigate the critical impact of the codebook size and report our findings below. We observed that using a smaller codebook size ($V=256$) combined with a weak regularization ($\epsilon=0.02$) for the noise-invariant learning caused the encoder to overfit, which we attribute to insufficient codebook utilization. To mitigate this, we increased the smoothing parameter to $\epsilon=0.05$ for the smaller codebook. This promoted more uniform codeword usage and successfully prevented collapse. Nevertheless, the model's performance with the smaller codebook still degraded compared to our final $V=2048$ setting.
>
>     | Pre-training Configuration | SIG | BAK | OVL | Spk Sim | WER |
>     | :--- | :--- | :--- | :--- | :--- | :--- |
>     | UNISE ($V=2048$) | **3.601** | **4.045** | **3.312** | **0.861** | **0.0761** |
>     | Small codebook ($V=256$) | 3.568 | 4.001 | 3.257 | 0.858 | 0.0902 |
>
>     We hypothesize that the larger codebook helps capture more fine-grained phoneme representations, which are crucial for this task. This observation aligns with the findings reported in [4].
>
> We add these details and the additional results in the revision (Appendix A and Section 3.4).
>
> **Closing.**
> We thank the reviewer again for the constructive feedback. All requested clarifications and additional analyses have been incorporated into the revision, and we welcome any further comments or questions.
>
> **References**
> - **[1]** Cho et al. Sylber: Syllabic embedding representation of speech from raw audio." ICLR 2025.
> - **[2]** Saijo et al., Interspeech 2025 URGENT speech enhancement challenge., Interspeech 2025.
> - **[3]** Caron et al.,Unsupervised Learning of Visual Features by Contrasting Cluster Assignments, NeurIPS 2020.
> - **[4]** Chang et al., Self-supervised Fine-tuning for Improved Content Representations by Speaker-invariant Clustering, Interspeech 2023.

---

### Official Review · Reviewer_vPdR · 2025-11-01

**Soundness:** 2
**Presentation:** 2
**Contribution:** 2
**Rating:** 2
**Confidence:** 4

**Summary:**

This paper proposes a unified generative speech enhancement method that aims to improve both acoustic quality and linguistic content preservation. The authors argue that prior approaches mainly emphasize perceptual quality while overlooking semantic consistency. To address this, they introduce an encoder trained with a contrastive loss to learn noise-invariant representations, coupled with a generative decoder that reconstructs enhanced speech from this bottleneck. Experimental results on several benchmarks show moderate improvements in content-preservation metrics. However, the evaluation is limited to speech enhancement and speech recognition tasks, which do not sufficiently demonstrate the generality of the proposed method. Moreover, the ASR performance on the CHiME-4 dataset remains notably below the mainstream average level.

**Strengths:**

The paper addresses an important and somewhat under-explored aspect of speech enhancement, content/semantic preservation rather than just clean sounding output. The framework is presented as unified representation learning + generation rather than simply plugging in a new decoder. The provided experimental results demonstrated the effectiveness of the proposed method.

**Weaknesses:**

1. The novelty appears limited. While combining contrastive representation learning with generative speech enhancement is interesting, the method largely builds upon existing methods without introducing a clear architectural or theoretical innovation.
2. The description of the noise-invariant encoder and the contrastive training setup is insufficiently detailed. It is unclear how noise invariance is explicitly enforced, how positive and negative pairs are constructed, and how this representation interacts with the generative decoder during training and inference.
3. The experimental validation needs improvement. It is unclear whether the reported improvements are statistically significant or robust across various corruption types and severities. The evaluation covers a narrow set of datasets and noise conditions, which raises concerns about the model’s generalization to unseen corruptions such as reverberation, clipping, or codec artifacts. The claim of enhanced content preservation is not well supported, as the ASR performance on the EARS test sets is even lower than that of the unprocessed noisy speech.

**Questions:**

1. Could the authors clarify the exact formulation of the contrastive loss? Specifically, how are the positive and negative pairs defined, e.g., clean vs. noisy, same vs. different speakers, or different noise types?
2. What types and severities of corruptions are considered in the evaluation?
3. How can “content preservation” be quantitatively measured? Are intelligibility metrics such as word error rate or human listening tests used to verify that linguistic content is better preserved?
4. How can the authors demonstrate that the observed gains originate from the proposed noise-invariant representation, rather than from increased model capacity or a stronger decoder architecture?

---

> ### Author Response · Authors · 2025-11-21
> **Response to Reviewer vPdR (1/3)**
>
> We thank Reviewer vPdR for the detailed and constructive review. We appreciate the positive feedback on the problem motivation and the importance of semantic preservation in speech enhancement. Below we address each concern in detail.
>
> **W1 -- Novelty and Conceptual Contribution.**
> We acknowledge the concern regarding novelty. Our contribution is a unified, **jointly training framework** for representation learning in speech enhancement (SE) that couples (i) self-supervised **noise-invariant clustering** and (ii) generative flow-based reconstruction.
> Unlike prior work on SE, which relies on existing pre-trained representations that are not noise-robust and focus primarily on discriminative tasks, our encoder and decoder are optimized together end-to-end under a unified objective, ensuring that the learned representations are noise-robust and aligned with the decoder’s reconstruction. This joint training allows the encoder to produce representations that are both noise-robust and well-suited for generative reconstruction, addressing limitations of previous methods that treat the encoder and decoder separately.
>
> **W2, Q1 -- Insufficient Description of Noise Invariance and Training Setup.**
> We apologize for the insufficient detail. To clarify, our method is more accurately described as noise-invariant clustering (based on SwAV [1]), not common contrastive learning. We include more details of self-supervised learning part in revision (Appendix A).
> Below are the answers to the reviewer's questions.
>
> 1. **How is noise invariance enforced?** We enforce invariance by taking one clean speech segment and creating two different noisy augmentations (views). The encoder is trained to map both of these different noisy inputs to the same cluster assignment (or "code"). This forces the encoder to learn representations that are invariant to noise and capture only the underlying speech content.
>
> 2. **How are positive and negative pairs constructed?** There are no explicit positive or negative pairs. The two random noisy augmentations (adding noise and reverberation) of the same utterance are treated as a positive pair.
>
> 3. **How does the encoder interact with the decoder?**
>
> *During pre-training:*
> The encoder and decoder are trained jointly in an end-to-end fashion. The encoder learns noise-invariant semantic representations through the explicit noise-invariant clustering objective, while the decoder simultaneously learns to reconstruct clean speech conditioned on these representations. The decoder’s reconstruction loss does encourage the encoder to produce features that are useful for the generative SE process. This joint optimization distinguishes our method from approaches that use a separately pre-trained, frozen encoder.
>
> *During fine-tuning on downstream tasks:*
> For downstream SE, we freeze the encoder and fine-tune only the decoder. While unfreezing the encoder is possible, our experiments show that freezing it preserves linguistic content more reliably, since updating it solely with the generative loss can degrade its noise-invariant semantic structure. We have added this result to the revision (Table 2).
>
> *During inference:*
> For speech enhancement, the noisy speech is passed through the encoder, whose noise-invariant representation conditions the decoder to generate the clean signal. The encoder can also serve as a standalone representation for discriminative downstream tasks such as noisy ASR.

---

> ### Author Response · Authors · 2025-11-21
> **Response to Reviewer vPdR (2/3)**
>
> **W3, Q2 -- Evaluation Scope, Corruption Diversity, and Content Preservation**
> We agree that a broader set of corruption conditions would further strengthen our work. Our current evaluation includes additive noise and reverberation, which are the most common conditions addressed by the baseline models. We made this choice to ensure a fair comparison with prior work. While a "universal" speech enhancement model that also handles clipping, codec artifacts (MP3/Opus), etc., is an important goal, this is often treated as a separate and more extensive task. Therefore, we left this extension as a valuable direction for future work.
>
> Regarding your comment on the EARS test sets, the WER values are higher than those of unprocessed noisy speech for all models. While enhancement often improves perceptual quality, the linguistic and speaker content can be altered, especially for harsh and highly diverse samples such as those in EARS. This issue is more pronounced in generative SE models, which reconstruct the waveform by sampling from a learned distribution. Unlike predictive (discriminative) models that directly estimate a deterministic mapping, generative models inherently allow more flexibility, which can lead to subtle shifts in phonetic details or speaker characteristics.
> This trend aligns with findings reported in the URGENT Challenge 2025 [2].
> Even so, our model achieves lower WER than other generative baselines, indicating that it preserves linguistic content more effectively. This suggests that our noise-invariant representation learning provides stronger semantic stability during reconstruction compared to prior generative approaches.
>
> **Q3 -- Measuring Content Preservation.**
> We primarily assess content preservation using word error rate (WER) in Table 2 and mutual information (MI) curves in Figure 4 of the appendix, both of which directly measure how well linguistic information is maintained.
> Recent studies [2,3,4] have already begun evaluating metrics beyond DNSMOS, such as WER or speaker similarity, to better capture content preservation.
> Our model achieves the best WER among generative SE methods and shows consistently high MI across different SNR levels, demonstrating robust preservation of linguistic content under varying noise conditions.

---

> ### Author Response · Authors · 2025-11-21
> **Response to Reviewer vPdR (3/3)**
>
> **Q4 -- Contribution of Encoder vs. Model Size.**
> This is a critical point. To isolate our encoder's contribution from overall model size, we performed a capacity-controlled ablation study (Section 3.4).
>
> Our results show that the observed gains are not simply due to model capacity. Other baselines, such as DiTSE [5], employ decoder with larger channels, yet our model still outperforms them in WER.
> In particular, when using the same decoder and training setup (Section 3.4 Effect of Representations on Speech Enhancement), representation from UNISE yields the strongest SE performance compared to alternative representations, such as WavLM with the same architecture, or a variant without any semantic representation. These results demonstrate that the improvements originate directly from our proposed method.
>
> Moreover, on the SE task from the SUPERB benchmark, our representation achieves superior performance compared to other SSL features (HuBERT and WavLM) under the same setup, further supporting that the gains stem from the representation rather than model capacity. This benchmark result is included in the revision (Appendix B.2).
>
> **Closing.**
> We thank the reviewer again for the helpful comments. In the revision, we have expanded methodological details and added further analyses to highlight the strengths of our model based on your comments. We welcome any additional comments or questions.
>
> **References**
> - **[1]** Caron et al.,Unsupervised Learning of Visual Features by Contrasting Cluster Assignments, NeurIPS 2020.
> - **[2]** Saijo et al., Interspeech 2025 URGENT speech enhancement challenge., Interspeech 2025.
> - **[3]** Yang et al., Genhancer: High-Fidelity Speech Enhancement via Generative Modeling on Discrete Codec Tokens, Interspeech 2024.
> - **[4]** Wang et al., FlowSE: Efficient and High-Quality Speech Enhancement via Flow Matching, Interspeech 2025.
> - **[5]** Guimarães et al., DiTSE: High-Fidelity Generative Speech Enhancement via Latent Diffusion Transformers, preprint.

---

### Official Review · Reviewer_Qwzv · 2025-11-03

**Soundness:** 2
**Presentation:** 3
**Contribution:** 2
**Rating:** 4
**Confidence:** 4

**Summary:**

The paper proposes UNISE (Unified Noise-Invariant learning for Speech Enhancement), which jointly trains (1) a contrastive, noise-invariant representation encoder (inspired by SwAV / swapped prediction with Sylber pseudo-labels) and (2) a Flow-Matching based generative decoder conditioned on the encoder’s latent to improve speech enhancement while better preserving linguistic/semantic content. Experiments are reported on multiple benchmarks (DNS-Challenge, VoiceBank-DEMAND, EARS, CHiME-4) comparing regression, diffusion, language-model-based, and other generative approaches. The authors show notable WER improvements and competitive perceptual metrics.

**Strengths:**

1. Clear motivation addressing semantic loss/hallucination in generative SE.
2. A joint training framework (encoder + conditional flow decoder) with supporting ablation studies.
3. Empirical improvements in WER while maintaining competitive perceptual metrics, suggesting a good trade-off between intelligibility and quality.

**Weaknesses:**

1. Reproducibility details missing: many hyperparameters and exact procedures (codebook training, Sinkhorn params, pseudo-label pipeline, training compute/time, seeds) are not fully specified.
2. Subjective evaluation lacks clarity: If human listening tests were conducted, their protocol is not sufficiently described; if not, the absence of blind subjective tests is a limitation.
3. Failure cases: Some benchmarks (e.g., CHiME-4) show that UNISE is not always best; the manuscript’s discussion of limitations and failure modes is brief.
4. Ambiguous effect of codebook / pseudo-label design choices. Key design parameters (codebook size, k-means clusters, pseudo-label source data) may strongly influence representation quality; the paper lacks comprehensive ablation experiments to validate its robustness.
5. No clear error analysis for cases where WER degrades or hallucinations occur.

**Questions:**

1. For the clustering/codebook and Sinkhorn steps: what were the exact hyperparameters (ε, number of iterations, temperature τ)? How was the collapse prevented? How sensitive is the model to the temperature or codebook size in contrastive learning?
2. Does joint optimization ever cause overfitting to certain noise types or degrade generalization to unseen environments?
3. Please detail the Sylber pseudo-label generation: which datasets were used to train the k-means, initialization method, how frame-level alignment was handled, and any preprocessing used. Would alternative pseudo-labels change results?
4. Why do you freeze the encoder during finetuning? What happens if the encoder is unfrozen for end-to-end finetuning—does performance improve or degrade? Please report experiments.
5. Please include more failure-case examples and a deeper analysis of when and why semantic hallucinations or distortions occur, and possible mitigation strategies.
5. Have the authors explored scaling (e.g., larger encoder or DiT-like decoder) and its effect on robustness?

---

> ### Author Response · Authors · 2025-11-21
> **Response to Reviewer Qwzv (1/4)**
>
> We thank Reviewer Qwzv for the constructive and detailed review. We appreciate the positive assessment of our motivation, the unified framework, and the empirical improvements on intelligibility while maintaining perceptual quality.
>
> Below we provide point-by-point responses, organized following the reviewer’s concerns. Several issues appear to stem from missing implementation details in the manuscript, and we welcome the opportunity to clarify and extend these, both here and in the revised version.
>
> **W1 -- Reproducibility and Missing Hyperparameters.**
> We agree that the manuscript omitted some training details, particularly regarding the encoder training procedure. We now provide the full specifications:
>
> **Noise-Invariant Clustering**
> Our approach is based on Swapping Assignments between Views (SwAV) [1].
> We added reproducibility details of the codebook training in Appendix A, including a more detailed explanation of the target distribution obtained via an entropy-regularized optimal transport problem:
>
> \begin{equation}
> Q^* = \arg\max_Q \; \mathrm{Tr}(Q C Z^\top) + \epsilon H(Q),
> \end{equation}
>
> where $C$ is the codebook, $Z$ are the feature representations, $H(Q)$ is the entropy function, and $\epsilon$ controls the smoothness of the distribution. The optimal $Q^*$ is efficiently computed using the Sinkhorn--Knopp algorithm (Appendix A).
> Please check the revision for more details.
>
> **Sinkhorn Parameters**
> Following Spin [2], we set the codebook size \(V=2048\), smoothness parameter to \(\epsilon=0.02\), iterates 3 times and a temperature of \(\tau=0.1\).
>
> **Pseudo-Label Pipeline**
> Full details of the pseudo-label generation process are provided in our response to Q3 and have been included in Section 3.1 of the revised manuscript.
>
> **Training Compute**
> Training is conducted on a single NVIDIA H200 GPU with a batch size corresponding to 300 seconds of audio per view (600 seconds total for the decoder input), for 200k steps. The full pre-training takes approximately 2.5 days. We include the full details in the revision.
>
> **W2 -- Subjective Evaluation Clarity.**
> We did not conduct formal human listening tests for this submission. Instead, we relied on objective evaluations, specifically DNSMOS for perceptual quality and Word Error Rate (WER) for intelligibility. As our research primarily focuses on content preservation, we placed a greater emphasis on WER, which is highly correlated with linguistic accuracy. The development of more appropriate metrics for evaluating content preservation remains an important area for future work.
>
> **W3 -- Failure Cases and Benchmark Weaknesses.**
> We acknowledge the reviewer's point regarding the CHiME-4 ASR benchmark. However, it is important to note that these results correspond to the performance of the pre-trained encoder representations on a noisy ASR task, one of several downstream discriminative tasks. Our aim was to demonstrate that our encoder can serve as a pre-trained representation applicable to various downstream tasks, not just SE. As mentioned in the limitations section of our conclusion, UNISE underperforms on this task.
>
> Our model primary focuses is on the speech enhancement (SE) task, rather than other downstream discriminative tasks like ASR. We observed that this focus can lead to a performance drop in recognition tasks. This is a known trade-off, and similar findings have been reported in related work; for instance, UniWav also noted a persistent gap in discriminative task performance at similar model scales. Nevertheless, our model still achieves competitive results on noisy ASR downstream tasks. We provide a brief discussion of this limitation in the conclusion to highlight this trade-off.

---

> ### Author Response · Authors · 2025-11-21
> **Response to Reviewer Qwzv (2/4)**
>
> **W4 -- Effect of Codebook & Pseudo-Label Choices.**
> We thank the reviewer for this constructive point. Due to computational constraints, we were unable to perform an exhaustive experimental sweep of all design choices. However, we did investigate the impact of the codebook size and report our findings below. We observed that using a smaller codebook size ($V=256$) combined with a weak regularization ($\epsilon=0.02$) for the noise-invariant learning caused the encoder to overfit, likely due to insufficient codebook utilization. To mitigate this, we increased the smoothing parameter to ($\epsilon=0.05$) for the smaller codebook, which successfully prevented collapse. Nevertheless, the model's performance with the smaller codebook still degraded compared to our final ($V=2048$) and ($\epsilon=0.02$) setting. DNSMOS and speaker similarity show relatively small gaps, while WER increases notably with the smaller codebook, indicating a loss in content preservation capability.
>
> **Pre-training Configuration**       | **SIG**  | **BAK**  | **OVL**  | **Spk Sim** | **WER**
> ------------------------------------|----------|----------|----------|-------------|--------
> UNISE ($V=2048$)  | **3.601** | **4.045** | **3.312** | **0.861**  | **0.0761**
> Small codebook ($V=256$) | 3.568   | 4.001   | 3.257   | 0.858      | 0.0902
>
> We hypothesize that the larger codebook is beneficial for capturing more fine-grained phoneme representations, which are crucial for the task. We add these results in the ablation section in the revision. This observation aligns with the findings reported in [2].
>
> Regarding the pseudo-label design, while we could not experiment with alternative pseudo-label types, our ablation study (as shown in Table 4, w/o $L_\text{Aux}$) did confirm that the inclusion of this auxiliary loss significantly improves content preservation, validating its role in our framework.
>
> **W5, Q5 -- More Failure Cases and Hallucination Analysis.**
> We added visualizations of examples in the revised manuscript, including comparisons with the baseline in Figure 3 and comparisons between ablated models in Appendix C. While it is difficult to specify all failure cases, these examples help readers understand the limitations of previous methods and the motivation for our model. In particular, by analyzing phoneme error rate (PER) versus DNSMOS on ablated models (Appendix B.5) and examining representative examples (Appendix C), we show that some samples are perceptually good but contain incorrect content in previous approaches, whereas our model preserves content accurately. Additional samples are already available on the demo page. Please refer to the revision for further details.

---

> ### Author Response · Authors · 2025-11-21
> **Response to Reviewer Qwzv (3/4)**
>
> **Q1 -- Sensitivity to Sinkhorn / Codebook Parameters.**
> As detailed in our response to W4, the sensitivity is directly linked to the codebook size. A small codebook ($V=256$) was sensitive to the smoothing parameter ($\epsilon=0.02$) and prone to collapse, requiring a higher $\epsilon=0.05$ for stability. In contrast, our final large codebook ($V=2048$) performed well even with weak regularization.
>
> **Q2 -- Joint Optimization and Generalization.**
> We tested our model's generalization capabilities by evaluating it on several diverse, unseen datasets, including challenging real-world recordings. As shown in our results tables, our model achieves consistently strong performance across these varied conditions. This suggests that our joint optimization strategy does not overfit to specific noise types and successfully generalizes to new acoustic environments.
>
> **Q3 -- Sylber Pseudo-Label Generation Details.**
> Sylber pseudo-label tokens are obtained by applying k-means clustering with 20k clusters to the segment-averaged Sylber [3] features extracted from the same training set used for our model. Clustering is performed using the MiniBatchKMeans implementation from the scikit-learn package, with a mini-batch size of 10,000 frames and the k-means++ initialization strategy with 5 random restarts. The resulting discrete cluster indices are then expanded back to match the durations of the original segments based on the frame alignments provided by the Sylber model, while non-speech regions are filled with a null token. No additional preprocessing is applied; we follow the official Sylber code. These details are included in Section 3.1 of the revised manuscript.
>
> Regarding the pseudo-label design, while we could not experiment with alternative pseudo-label types, our ablation study (as shown in Table 4) did confirm that the inclusion of this auxiliary loss significantly improves content preservation, validating its role in our framework. We chose Sylber features because they demonstrate superior performance compared to common representations (e.g., HuBERT) in resynthesis and provide a compact, targeted pseudo-label signal as a weak auxiliary. However, investigating alternatives, such as phonetic-, word-, or syllabic-level pseudo-labels like ours, represents a valuable direction for future work.

---

> ### Author Response · Authors · 2025-11-21
> **Response to Reviewer Qwzv (4/4)**
>
> **Q4 -- Why Freezing the Encoder During Finetuning?**
> Our initial approach was to freeze the encoder, treating it as a fixed semantic feature extractor for content, consistent with prior works in SE.
>
> However, we agree with the reviewer's point that the encoder could also be used as a strong initialization for end-to-end finetuning. We ran this experiment and found that unfreezing the encoder during finetuning led to an increase in WER (i.e., worse content preservation) compared to our original method (Section 3.2 Experimental Results and Table 2).
> We hypothesize this is because, without the constraint of the frozen weights, the encoder's parameters drift. The model begins to optimize purely for the acoustic aspects of the enhancement task using only the flow-matching (generation) loss, losing the explicit content preservation focus it was pre-trained for. Freezing the encoder ensures that this content preservation is maintained. We add this new ablation result and discussion in the revision.
>
> **Q6 -- Scaling the Model.**
> Due to computational power constraints, we were unable to conduct dedicated scaling experiments for this submission.
> However, we agree that this is a critical direction. Substantial evidence in the literature suggests that our framework would benefit significantly from scaling. For instance, UniWav [4] demonstrated the criticality of a larger encoder for the performance of their unified framework. Similarly, DiTSE [5] reported substantial improvements in speech enhancement, particularly in WER, when scaling the semantic feature model.
>
> Based on this, we strongly believe that scaling our model, for instance by increasing the number of channels and layers in both the encoder and decoder, would yield significant further improvements. We have noted this as an important direction for future work.
>
> **Closing Remarks.**
> We thank the reviewer again for the detailed and constructive comments. We have revised the manuscript to improve reproducibility, clarify the pseudo-label methodology, and expand the discussion of limitations and failure cases. We welcome any further comments or questions.
>
> **References**
> - **[1]** Caron et al., *Unsupervised Learning of Visual Features by Contrasting Cluster Assignments*, NeurIPS 2020.
> - **[2]** Chang et al., *Self-supervised Fine-tuning for Improved Content Representations by Speaker-invariant Clustering*, Interspeech 2023.
> - **[3]** Cho et al., *Sylber: Syllabic embedding representation of speech from raw audio*, ICLR 2025.
> - **[4]** Liu et al., *UniWav: Towards Unified Pre-training for Speech Representation Learning and Generation*, ICLR 2025.
> - **[5]** Guimarães et al., *DiTSE: High-Fidelity Generative Speech Enhancement via Latent Diffusion Transformers*, preprint.

---

### Author Response · Authors · 2025-11-21
**Summary of the Revision upon Feedback (1/2)**

Dear Reviewers,

We sincerely thank you all for your careful reading and constructive feedback. We have revised our manuscript to address all concerns raised, clarified methodological details, and added additional experiments and ablations. A highlighted PDF indicating all changes has also been uploaded.

Here is a summary of the major changes:

- **Reproducibility and Implementation Details:** We added full specifications for the noise-invariant encoder training, including a more detailed explanation of the optimal transport problem for target distributions and Sinkhorn parameters (Appendix A). Additionally, we provide details on the pseudo-label generation pipeline and training compute/time. These are described in Section 3.1 and elaborated in the revised manuscript.

- **Example Samples and In-Depth Result Analysis:** We expanded the discussion with additional analysis and examples of cases with high WER. We include additional visualizations with example samples in the revision for deeper analysis (Figures 3 and Appendix C), and provide a detailed examination of results on the particularly challenging EARS dataset (Section 3.2 and Appendix B.1). Furthermore, we analyze perceptual (DNSMOS) versus linguistic (PER) trade-offs, highlighting occurrences of semantic hallucinations in previous approaches (Appendix B.5).

- **Codebook Size Analysis:** We conducted an ablation study on a smaller codebook size to justify our choice of codebook size \(V=2048\) and entropy regularization parameter \(\epsilon=0.02\) (Section 3.4 Ablation on the Pre-training Method).

- **Additional Experiments:** We added an ablation experiment without SSL features (Section 3.4 Effect of Representations on Speech Enhancement) and benchmark results on SUPERB SE to demonstrate UNISE’s performance compared to other representations under a fair comparison (Appendix B.2).

- **Notation, Figures, and Presentation:** All inconsistencies in symbols, figure labels, and equations were corrected. The appendix is now referenced in the main text for context.

Minor changes for overall clarity.

We thank the reviewers again and look forward to your feedback!
We welcome any further comments or questions.

Best regards,
Authors

---

### Author Response · Authors · 2025-11-27
**Summary of the Revision upon Feedback (2/2)**

Dear Reviewers,

We are submitting a new revision adding deeper analysis on experimental results in the main text, where the previous revision had only included this analysis in Appendix B.1. We revised the content to provide more clarity by explicitly separating baselines into groups of deterministic and generative models in Speech Enhancement (SE) and explaining the results based on this division more explicitly. This is expected to ensure that the purpose of our work and the experimental results are clearly understood.

The following structural and content changes have been implemented to enhance clarity:
* **Table Purpose:** We explicitly state that Table 2 is a system-level comparison and have clarified the distinct roles of the main results (Table 2) and the controlled ablation (Table 5) throughout the text.
* **Model Categorization in Table 2:** We have explicitly categorized baselines as Discriminative (D) or Generative (G) within the table structure to improve comparison clarity. We also refined the list of deterministic baselines (adding DEMUCS/FRCRN) to provide stronger evidence.
* **Deeper Result Discussion:** The deeper discussion regarding the trade-offs inherent in generative SE models has been explained in the main text also, using Figure 3 as visual example, to ensure easy understanding of the results for the reader.

We kindly ask reviewers to refer to the revised experimental results section (Section 3.2), which now contains the integrated discussion and analysis critical for understanding the purpose and results of our work.

Best regards,
Authors

---

### Author Response · Authors · 2025-12-02
**Summary to Area Chair: Core Contributions & Key Updates during Discussion**

Dear Area Chair,

Thank you for overseeing our submission. Due to the unexpected situation at ICLR 2026, reviewers were unable to continue the rebuttal discussion. We summarize the key updates below. In response to the constructive feedback from Reviewers, we focused on **refining the presentation of our comparative analysis** and **strengthening the empirical validation** of our framework.

**Paper Contributions**

* **Unified Framework:** We propose UNISE, a unified framework that jointly trains a noise-invariant encoder (via clustering) and a flow-matching decoder. This framework bridges the gap between discriminative SSL and generative SE, yielding a representation that is robust to noise while retaining semantic content.
* **Content Preservation:** We address the critical issue of semantic hallucination in generative SE. By leveraging noise-invariant clustering and auxiliary pseudo-label losses, UNISE achieves superior content preservation (WER) compared to generative baselines.
* **Empirical Performance:** We demonstrate that our method achieves a favorable trade-off, securing the best intelligibility scores among generative models while maintaining competitive perceptual quality across multiple benchmarks.

**Key Updates**

**Validation of Design Choices \& Robustness**

* **Capacity-Controlled Ablation:** We highlighted that our existing ablation study (Table 5) specifically isolates the encoder's contribution by fixing the decoder capacity. We clarified in the text that this confirms UNISE's gains stem from the representation learning rather than model size.
* **Codebook \& Training Strategy:** We provided additional ablation studies to empirically justify our selected codebook size and fine-tuning strategy (freezing the encoder), confirming these choices are optimal for preserving semantic structure.
* **Extended Benchmarks:** We added results on the SUPERB SE benchmark as supplementary evidence of the representation's ability.
* **Visual Analysis:** We added illustrative examples (Figure 3 and Appendix C) to visualize the semantic hallucinations common in prior methods and demonstrate UNISE’s robustness against them.

**Clarification of Results \& Reproducibility**
* **Explicit Categorization:** To improve the readability of our main results (Table 2), we added explicit labels distinguishing **Discriminative (D)** from **Generative (G)** models. This clarifies performance trends, such as WER rank orders on challenging datasets.
* **Analysis of Generative Trade-offs:** We integrated a comprehensive discussion into the main text (Section 3.2) regarding the inherent trade-off between perceptual quality and linguistic fidelity in generative SE. We explicitly explain why generative models can suffer from phonetic drift (degrading WER) in harsh conditions compared to deterministic discriminative models. This analysis provides essential context for interpreting the results, highlighting that while the trade-off exists, UNISE achieves the highest stability among generative baselines.
* **Implementation Details:** We provided expanded details on the noise-invariant encoder training (including the entropy-regularized optimal transport formulation) and the Sylber pseudo-label generation pipeline to ensure full reproducibility.

**Summary**
Our revisions have improved the clarity and completeness of the manuscript. By explicitly distinguishing generative from discriminative baselines and analyzing the trade-offs between perceptual quality and linguistic fidelity, we have better aligned our results with the paper's core motivation. Supported by deeper discussion and additional evidence, the paper now clearly communicates its dual contribution: **a self-supervised noise-invariant representation for SE** and a following **content-preserving generative speech enhancement framework.**

---

### Meta-Review · Area_Chair_1tK2 · 2026-01-07

**Summary:**

The primary concerns are around the reproducibility details, the validity of the experimental comparisons (specifically regarding model size and fairness), and the justification of the trade-off between semantic for WER and perceptual quality (DNSMOS). There's concern on novelty of the proposed unified framework with existing techniques.

**Reviewer Concerns:**

Qwzv:
* reproducibility (missing hyperparameters, training config, pipeline information): the authors added substantial material in appendix to cover these, it would be better to have a concise version of that in the main paper.
* lack of human subjective eval:

vPdR:
* novelty, the unified framework only combines multiple known techniques: partially addressed as the author argues the "joint training" is the core novelty
* clarity on noise-invariance encoder and negative/positive pair construction: the authors added more details on those

zNt5:
* tradeoff between WER and perceptual quality: the authors provided results to explicitly discuss the tension between generative hallucination and content preservation
* details on codebook initialization, optimal transport params: the authors added the details

rWcK:
* math notation errors: addressed in revision
* model sizes not listed, comparisons unfair: added a base-sized model results
* tradeoff between WER and perceptual quality: same as above

**Reviewer Scores:**

Qwzv: 4, may change to 5
vPdR: 2, likely maintain
zNt5: 4, likely no change
rWck: 4, may change to 5

---

### Decision · Program_Chairs · 2026-01-26

Reject